# Increased iron uptake by splenic hematopoietic stem cells promotes TET2-dependent erythroid regeneration

Yu-Jung Tseng [1,8], Yuki Kageyama[2,8], Rebecca L. Murdaugh[3], Ayumi Kitano[2], Jong Hwan Kim[4], Kevin A. Hoegenauer[2], Jonathan Tiessen[3], Mackenzie H. Smith[2], Hidetaka Uryu [5,6], Koichi Takahashi [5,6], James F. Martin [4,7], Md Abul Hassan Samee [4] & Daisuke Nakada [1,2,3] ✉

Hematopoietic stem cells (HSCs) are capable of regenerating the blood system, but the instructive cues that direct HSCs to regenerate particular lineages lost to the injury remain elusive. Here, we show that iron is increasingly taken up by HSCs during anemia and induces erythroid gene expression and regeneration in a *Tet2*-dependent manner. Lineage tracing of HSCs reveals that HSCs respond to hemolytic anemia by increasing erythroid output. The number of HSCs in the spleen, but not bone marrow, increases upon anemia and these HSCs exhibit enhanced proliferation, erythroid differentiation, iron uptake, and TET2 protein expression. Increased iron in HSCs promotes DNA demethylation and expression of erythroid genes. Suppressing iron uptake or TET2 expression impairs erythroid genes expression and erythroid differentiation of HSCs; iron supplementation, however, augments these processes. These results establish that the physiological level of iron taken up by HSCs has an instructive role in promoting erythroid-biased differentiation of HSCs.

Acute hemolytic anemia is a common comorbidity of cancer, infection, and certain medications, and negatively impacts patient quality of life and treatment responses[1]. Maintenance of the hematopoietic system by hematopoietic stem cells (HSCs) is a vital physiological mechanism with vast applications for therapeutic intervention in hematopoietic disorders like anemia. However, our incomplete understanding of the pathobiology of HSCs has limited the development of treatments for this and other diseases.

Stress erythropoiesis is the process by which animals respond to anemic stress and generate a bolus of erythrocytes. Several characteristics distinguish stress from steady-state erythropoiesis. In adult mice, steady state erythropoiesis occurs in the bone marrow, while stress erythropoiesis occurs extramedullary in the spleen and liver[2,3]. Stress erythropoiesis also requires distinct signaling pathways from normal erythropoiesis, including those dependent on bone morphogenetic protein 4 (BMP4) and hedgehog[4,5]. A study using the *flexed-tail* mutant mice revealed that splenic megakaryocyte-erythroid progenitors (MEPs) are the BMP4-responsive stress erythroid progenitors capable of generating erythroid burst forming units (BFU-E) in vitro[6]. Subsequent studies found CD34+lin-Sca-1+c-kit+ cells, which are mostly composed of multipotent progenitor cells (MPPs) and hematopoietic progenitor cell 1 and 2 (HPC1 and 2)[7,8], as the immature progenitor that migrates from the bone marrow to the spleen in response to anemia and produce BMP-responsive BFU-E[4,9]. The study showed that HSCs

[1]Graduate Program in Translational Biology and Molecular Medicine, Baylor College of Medicine, Houston, TX 77030, USA. [2]Department of Molecular and Human Genetics, Baylor College of Medicine, Houston, TX 77030, USA. [3]Graduate Program in Developmental Biology, Baylor College of Medicine, Houston, TX 77030, USA. [4]Department of Integrative Physiology, Baylor College of Medicine, Houston, TX 77030, USA. [5]Department of Leukemia, The University of Texas MD Anderson Cancer Center, Houston, TX 77030, USA. [6]Department of Genomic Medicine, The University of Texas MD Anderson Cancer Center, Houston, TX 77030, USA. [7]Cardiomyocyte Renewal Laboratory, Texas Heart Institute, Houston, TX 77030, USA. [8]These authors contributed equally: Yu-Jung Tseng, Yuki Kageyama. ✉e-mail: nakada@bcm.edu

formed fewer BFU-E compared to MPPs upon transplantation, but it remained unclear whether HSCs are activated by anemia or whether they are involved in stress erythropoiesis[9]. Other studies, however, have indicated that HSCs (CD150+CD48-lin-Sca-1+c-kit+ cells) migrate to the spleen and respond to anemia caused by EPO overexpression, blood loss, or pregnancy[10–13]. The overall role of HSCs in the spleen in response to anemia remains unclear.

Iron plays a crucial role in both steady-state and hemolytic anemia-induced stress erythropoiesis. Increased iron availability resulting from hepcidin suppression during stress erythropoiesis leads to an approximately tenfold increase in iron uptake to support emergency hemoglobin synthesis[14,15]. Despite the confirmed role of iron on erythroid maturation, its direct effects on HSCs remain rather controversial. Iron overload impairs hematopoietic stem and progenitor cells (HSPCs) through iron-induced reactive oxygen species (ROS) and hinders HSC transplantation[16,17]. Depletion of F-box and leucine-rich repeat protein 5 (FBXL5), a regulator of iron homeostasis, disrupts the FBXL5-iron regulatory protein 2 (IRP2) axis, attenuating HSC function due to iron overload[18]. Iron insufficiency also negatively affects HSPC function and can lead to medical conditions, such as iron-deficiency anemia[19]. Loss of transferrin receptor 1 (Tfrc) reduced cellular iron content and hampered proliferation and differentiation of HSPCs, defects that were rescued by supplementing exogenous iron[20]. Together these studies show both pathological iron overload and insufficiency impair HSC function, but whether transient fluctuations in iron levels affect HSC behavior remains to be established.

TET family proteins are iron(II)/α-ketoglutarate (Fe(II)/α-KG)-dependent dioxygenases that oxidize 5-methyl-cytosine (5mC) to 5-hydroxymethyl-cytosine (5hmC), leading to DNA demethylation[21]. Among the TET proteins, the role of TET2 is most established in hematology, having been studied extensively in the context of clonal hematopoiesis and hematological malignancies[22–26]. Tet2 knockout HSCs exhibit transcriptional and differentiation skewing towards the myeloid lineage at the expense of erythroid lineage[27–29]. In humans, TET2 is expressed in erythroid cells and its loss results in defective erythroid development[29–31]. Supplementation of ascorbate, a cofactor of TET proteins that reduces Fe(III) to Fe(II)[32–34], promotes DNA demethylation and attenuates leukemia progression. On the other hand, ascorbate depletion reduces TET2 function and promotes HSC function and leukemogenesis[35,36]. Together, these studies suggest that cofactors fine tune TET2 activity in HSCs in order to balance differentiation and regeneration.

Here, we investigated the effects of anemic stress on HSCs. During phenylhydrazine (PHZ)-induced acute hemolytic anemia, the number of HSCs in the spleen increases and these cells exhibit enhanced proliferation, regeneration, and erythroid differentiation. These changes in splenic HSCs were associated with elevated intracellular iron levels, increased TET2 protein, and increased expression and DNA demethylation of erythropoiesis genes. Blocking iron uptake impaired splenic HSC expansion, proliferation, and erythroid potential. Suppressing TET2 expression also decreased the erythroid potential of splenic HSCs. Overall, our results provide insight into the instructive role of iron in promoting TET2-mediated DNA demethylation and erythropoiesis in splenic HSCs during anemia.

## Results

### PHZ-induced acute hemolytic anemia increases splenic HSCs
Recent studies using clonal barcoding and lineage tracing of HSCs in situ demonstrated that hematopoietic contribution by HSCs is an overall slow process[37–42]. T- and B-cells had particularly slow influx from HSCs, whereas platelets and granulomonocytic cells had more robust influx from HSCs. Applying stressors to these HSC tracing models revealed that the contribution of HSCs to multiple hematopoietic lineages increased after myeloablation or severe inflammation[39,41]. In contrast to these stressors that require

multilineage responses to return to homeostasis, hemolytic anemia leads to specific depletion of erythrocytes. Whether multipotent HSCs, in addition to lineage restricted erythroid progenitors, respond to such insult remains unknown. To investigate whether HSCs respond to hemolytic anemia, we used a HSC lineage tracing model with the Krt18-CreER driver (in Krt18-CreER; Rosa26-lox-STOP-lox-tdTomato mice, hereafter Krt18-tdT mice). In these mice, Krt18 is specifically expressed in HSCs and allows genetic labeling of HSCs and their descendants (Supplementary Fig. 1A)[41]. Recent analysis revealed that Krt18-CreER labels primitive HSCs as specifically as the Tie2-ER-Cre-ER driver[38,43]. Krt18-tdT mice were treated with tamoxifen for 5 consecutive days followed by two doses of phenylhydrazine (PHZ) to induce hemolytic anemia (Fig. 1a). Mice became anemic as soon as 1 day after PHZ treatment and recovered over the next 4 days (Fig. 1b). Spleens were enlarged in anemic mice and reached a maximum size around day 3–4 post PHZ treatment, returning to normal at day 7 (Supplementary Fig. 1B). After PHZ treatment, red blood cells (RBCs) were the only population within mature hematopoietic cells in the blood that exhibited a significant increase in tdTomato labeling compared to controls, indicating accelerated generation of RBCs from HSCs (Fig. 1d). In contrast to the multilineage response caused by chemotherapy 5-fluorouracil[41], Krt18-CreER-mediated labeling of other lineages was unaffected by PHZ treatment (Fig. 1d). Labeling in hematopoietic stem and progenitor cells (HSPCs) in the bone marrow was not affected by PHZ (Fig. 1c). However, MPPs and megakaryocyte-erythroid progenitors (MEPs) in the spleen exhibited increased labeling, indicative of enhanced erythroid regeneration in the spleen (Fig. 1e). These results suggest that HSCs respond to anemic stress by compensating for loss of RBCs without altering their contribution toward other lineages.

We then examined the effect of PHZ-induced hemolytic anemia on HSPCs. PHZ treatment reduced the number of HPC2 cells, which are biased towards the megakaryocytic and erythroid lineages[8,44], but did not affect the number of HSCs or other HSPCs in the bone marrow (Fig. 1f and Supplementary Fig. 1C, D). However, PHZ-treated mice had significantly expanded HSC and HPC2 numbers in their spleens, with peak expansion at 3 or 4 days after PHZ treatment, respectively, when RBCs also reached nadir (Fig. 1g and Supplementary Fig. 1D).

To determine the frequency of HSCs in a functional assay, we performed limiting dilution transplantation of whole bone marrow cells and splenocytes from mice treated or untreated with PHZ. This assay revealed that while PHZ treatment did not affect the frequency of HSCs in bone marrow (control; 1 in 35,555, PHZ; 1 in 52,023, $P = 0.485$), it increased HSC frequency in spleens fivefold (control; 1 in 3,067,740, PHZ; 1 in 618,953, $P = 0.0072$, Fig. 1h). These results establish that HSCs expand in the spleen following hemolytic anemia, with a concomitant increase in progenitors and cell populations to support increased erythropoiesis.

### Hemolytic anemia promotes erythropoiesis by splenic HSCs
To study how hemolytic anemia affects HSPCs, we first analyzed HSPC proliferation. We injected C57BL/6 mice with two doses of PHZ followed by a 24-h pulse of BrdU three days after the first PHZ injection. We found significantly higher BrdU incorporation into splenic but not bone marrow HSCs and MPPs after PHZ treatment compared to control, indicating an enhanced cell division in these two splenic populations (Fig. 2a). We then examined the differentiation potential of HSCs in single cell colony forming assays. While splenic HSCs from control mice formed significantly fewer colonies than bone marrow HSCs, splenic HSCs from PHZ-treated mice formed significantly more colonies than those from untreated mice (Fig. 2b). By enumerating the types of colonies formed by single HSCs, we found that PHZ-treated splenic HSCs formed more multipotent colonies consisting of granulocytes, erythrocytes, megakaryocytes, and macrophages/monocytes (GEMm) than control splenic HSCs (Fig. 2c and Supplementary Fig. 2A).

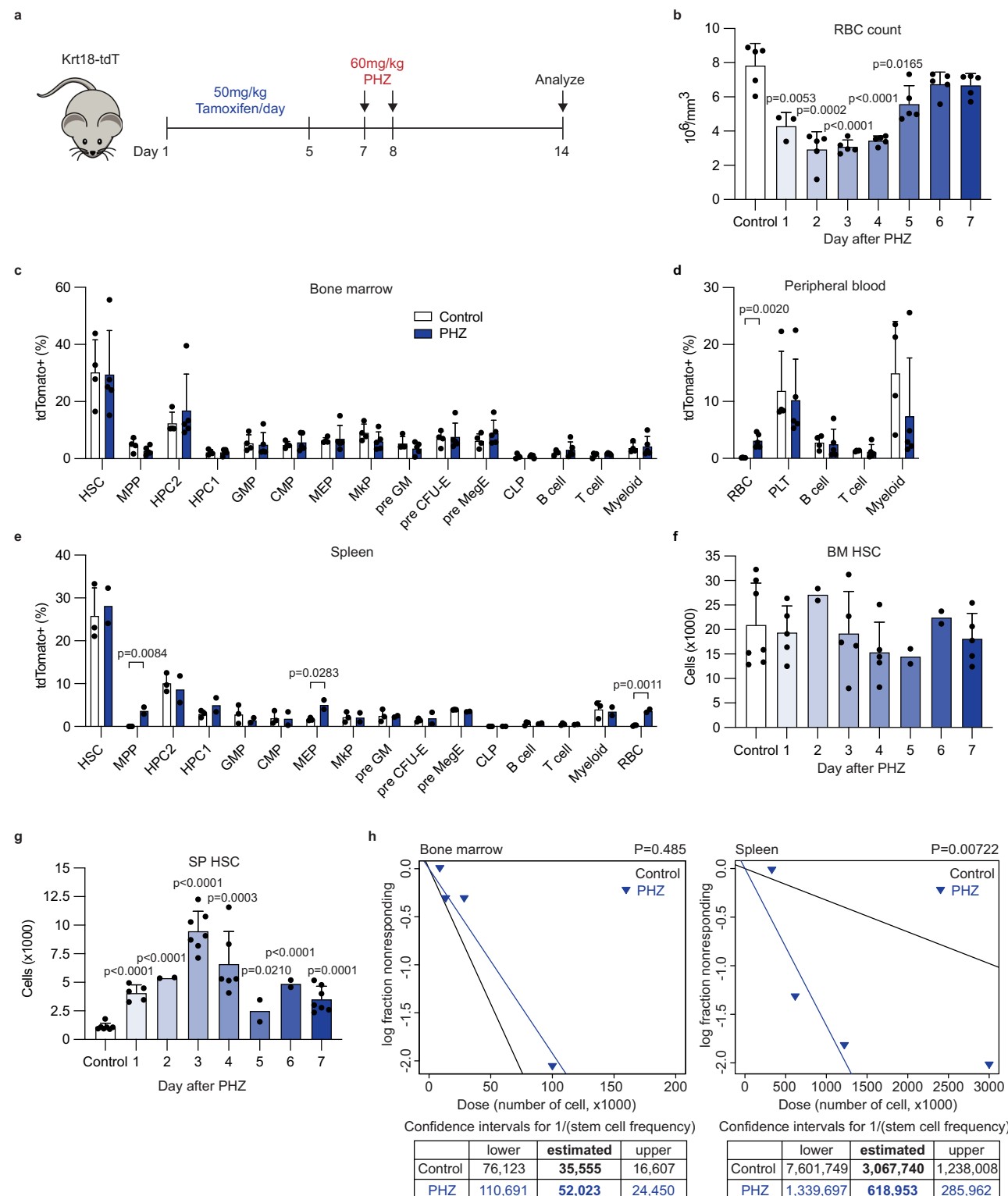

**Fig. 1 | Splenic HSCs are increased in response to PHZ-induced hemolytic anemia. a** Schematic illustration of the HSC lineage tracing experiment. **b** Red blood cell (RBC) count at days 1–7 post PHZ treatment (*n* = 3 for day 1 and *n* = 5 for others). **c**–**e** Labeling frequencies of different hematopoietic lineages in the bone marrow and peripheral blood (**c**, **d**) or in the spleen (**e**) of PHZ-treated (blue bars) or untreated (open bars) Krt18-tdT mice (*n* = 4, 5, 4, 5, 3, and 2 for untreated bone marrow, PHZ-treated bone marrow, untreated peripheral blood, PHZ-treated peripheral blood, untreated spleen, and PHZ-treated spleen, respectively). **f**, **g** The numbers of HSCs in the bone marrow (**f**, BM HSC) and spleen (**g**, SP HSC) at days 1–7 post PHZ treatment (*n* = 7, 5, 2, 5, 5, 2, 2, and 5 for control and days 1–7, respectively

in (**f**); *n* = 7, 5, 2, 7, 6, 2, 2, and 7 for control and days 1–7, respectively in (**g**)). **h** Limiting dilution transplantation of bone marrow cells (left) or splenocytes (right) isolated from control (black) or PHZ-treated (blue) mice. Estimated HSC frequencies and 95% confidence intervals are shown. Comparisons were performed by unpaired, two-tailed Student's *t* test in (**b**–**g**) and Poisson statistical analysis in (**h**). All data represent mean ± standard deviation. All numbers (*n*) are independent animals. Source data are provided as a Source Data file. Mouse illustration was adapted from Creazilla (https://creazilla.com/nodes/18581-big-eared-grey-mouse-clipart) under the Creative Commons license CC0.

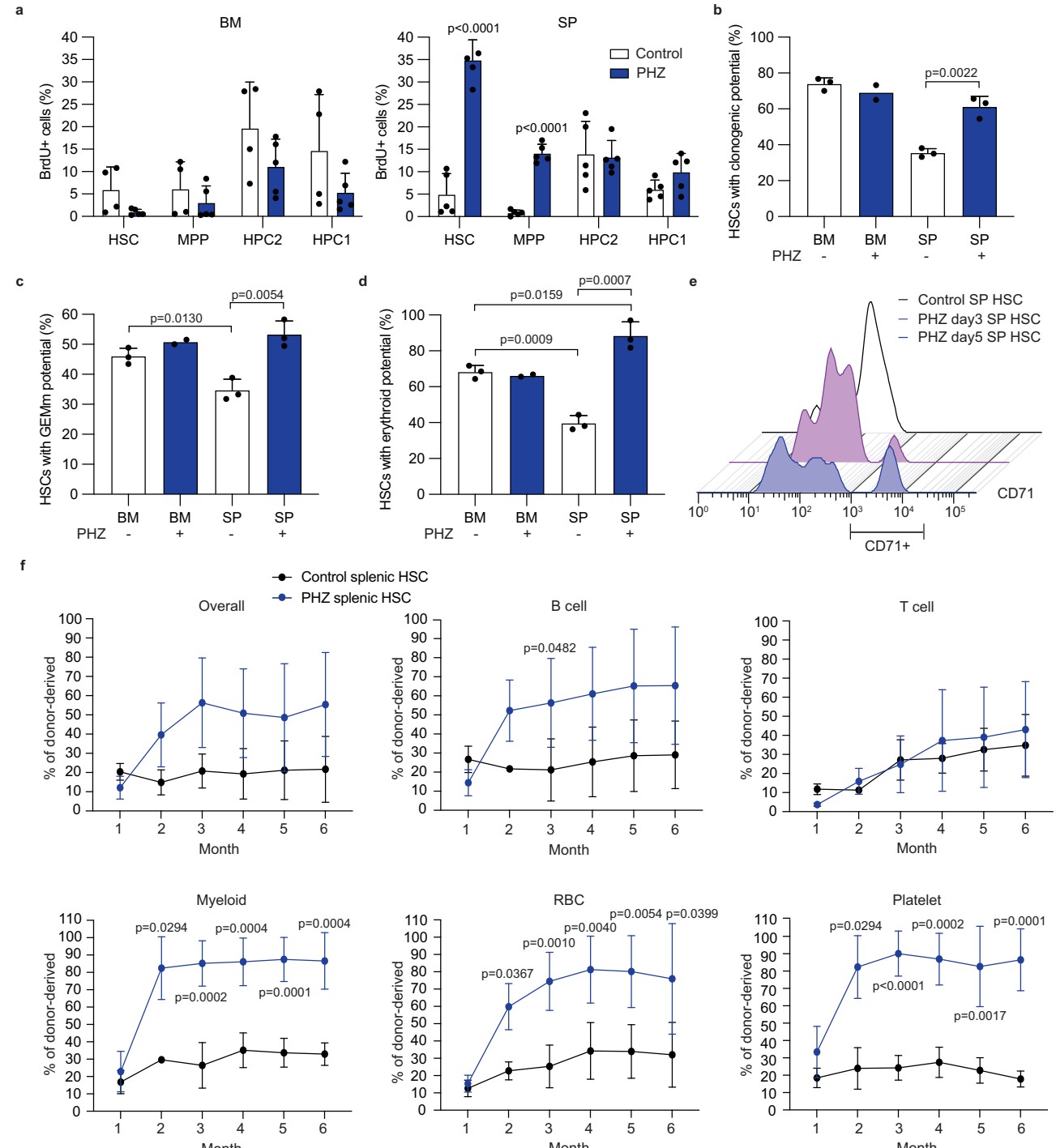

**Fig. 2 | Hemolytic anemia increases regenerative and erythroid potential of splenic HSCs. a** BrdU incorporation assay with bone marrow (BM, left) and splenic (SP, right) HSC, MPP, HPC2 and HPC1 from mice treated or untreated with PHZ ($n = 4$ for control BM and $n = 5$ for other groups). **b–d** Colony forming assays with single-cell sorted bone marrow or splenic HSCs from either control or PHZ-treated mice. Shown are clonality of single HSCs (**b**), the frequency of colonies containing granulocytes, erythrocytes, megakaryocytes, and macrophages/monocytes (GEMm, **c**), and those containing erythroid cells (**d**) ($n = 2$ for PHZ-treated bone marrow and $n = 3$ for other groups). **e** Representative histograms of CD71 expression in splenic HSCs at the indicated time point after PHZ treatment. **f** Transplantation of 50 splenic HSCs isolated from mice treated or untreated with PHZ. Proportion of donor-derived cells in each cell population are shown ($n = 3$ for control and $n = 4$ for PHZ in T cell; $n = 4$ for control and $n = 5$ for PHZ in other groups). Comparisons were performed by unpaired, two-tailed Student's $t$ test. All data represent mean ± standard deviation. All numbers ($n$) are independent animals. Source data are provided as a Source Data file.

The frequency of colonies that contained Ter119+ erythroid cells was significantly lower in control splenic HSCs compared to bone marrow HSCs, but PHZ treatment significantly increased these colony numbers to levels higher than control bone marrow HSCs (Fig. 2d). Notably, a subset of splenic HSCs expressed CD71, the transferrin receptor enriched in erythroid cells, after PHZ-induced anemia (Fig. 2e and Supplementary Fig. 2B).

We then tested whether hemolytic anemia affects the regenerative potential of HSCs in vivo. We sorted 50 bone marrow or splenic HSCs from control or PHZ-treated *Ubc-GFP* mice and transplanted

them to lethally irradiated recipients. Splenic HSCs exhibited lower reconstitution compared to bone marrow HSCs from control mice (Supplementary Fig. 2C). However, splenic HSCs isolated from PHZ-treated mice showed significantly higher reconstitution of RBCs, myeloid cells, and platelets compared to control splenic HSCs (Fig. 2f). Additionally, PHZ-treated splenic HSCs exhibited increased potential to generate stress erythroid progenitors[9] in the spleens after transplantation compared to control splenic HSCs (Supplementary Fig. 2D). PHZ treatment did not affect the reconstitution potential of bone marrow HSCs (Supplementary Fig. 2C). Collectively, these results demonstrate that PHZ-induced hemolytic anemia increases the regenerative potential of splenic HSCs, especially in the erythro-myeloid lineage.

To explore the changes in the HSPC population caused by hemolytic anemia, we performed single cell RNA sequencing (scRNA-seq) of lineage⁻Sca-1⁺c-kit⁺ (LSK) cells isolated from the bone marrow or the spleens of mice treated or untreated with PHZ. We identified twelve clusters including those that are enriched in genes associated with HSC, myeloerythroid progenitor cells (MPP-My/Ery), and erythroblasts (EB) (Fig. 3a, b). Consistent with the expansion of HPC2 and the increased erythropoiesis potential of the splenic progenitors after PHZ treatment, we observed significant expansion of clusters MPP-My/Ery and EB in the spleens of PHZ-treated mice (Fig. 3c, d). Moreover, cluster EB was uniquely present in the spleens of PHZ-treated mice (Fig. 3c).

Cluster MPP-My/Ery could be further subdivided into three clusters, namely SC1, SC2, and SC3 (Fig. 3e). Cluster SC1 exhibited mix erythroid and myeloid potential, cluster SC2 showed a stronger bias towards erythroid potential, and cluster SC3 demonstrated a greater commitment to the myeloid lineage (Supplementary Fig. 2E). We then thought to delineate the developmental trajectories between immature HSC and the myeloerythroid progenitors (clusters SC1, SC2, SC3, and EB). We observed a bifurcation in the trajectory from HSC towards either cluster SC1 or SC2 (Fig. 3f). The trajectory towards cluster SC1 was prominent in the bone marrow and in the control spleen, whereas PHZ-treated spleens exhibited strong enrichment of cells following the trajectory towards cluster SC2 and then to cluster EB, the erythroblasts (Fig. 3g). These findings indicate that PHZ-induced hemolytic anemia promotes erythroid commitment in splenic HSPCs and expands progenitors with erythroid potential.

### Iron promotes erythroid differentiation of splenic HSCs

Stress erythropoiesis is associated with increased iron consumption due to enhanced hemoglobin synthesis[45]. PHZ-induced hemolytic anemia increases levels of splenic iron and iron absorption[46], but whether iron uptake is increased in splenic HSCs remains unknown. To test this, we measured iron levels in HSCs with calcein AM. Fluorescence of calcein AM is quenched by the cellular labile iron pool, and therefore is anticorrelated with cellular iron level[47,48]. As soon as one day after PHZ treatment, the iron level in splenic HSCs, but not bone marrow HSCs, was significantly increased, reaching the highest level around day 3 post treatment before returning to steady-state at day 7 (Fig. 4a, b). Daily treatment with an iron chelator deferoxamine (DFO) reduced PHZ-induced splenic HSC iron uptake (Fig. 4c). Importantly, disrupting iron uptake by DFO partially diminished the PHZ-induced splenic HSC expansion and proliferation (Fig. 4d, e). DFO treatment also suppressed the enhanced colony forming capacity and erythroid potential of PHZ-treated splenic HSCs (Fig. 4f–h and Supplementary Fig. 3A). These results demonstrate not only that HSCs take up more iron following PHZ treatment, but also that this increase in iron promotes HSC expansion, division, and supports the enhanced erythroid potential observed during anemia.

We used two different models of erythropoietic stress, namely repeated bleeding (Fig. 4i)[13] and transfusion of heat-stress erythrocytes (Supplementary Fig. 3B)[49,50], to examine whether the changes in

splenic HSCs upon erythroid stress is specific to PHZ treatment. Repeated bleeding increased splenic but not bone marrow HSC numbers, which was accompanied by increased iron uptake by splenic HSCs (Fig. 4j, k). These changes were partially rescued by RBC transfusion (Fig. 4j, k). Transfusion of heat-stressed erythrocytes models pathologically shortened erythrocyte lifespan[49,50]. Two hours after transfusion of stressed erythrocytes, but not fresh erythrocytes, we observed slightly enlarged and darkened spleens as in PHZ-treated animals, indicating enhanced erythrophagocytosis in the spleen (Supplementary Fig. 3B, C). Transfusion of stressed erythrocytes increased splenic HSCs, although to a much lower degree than PHZ treatment, and increased iron uptake of splenic HSCs without affecting bone marrow HSCs (Supplementary Fig. 3D–F). These results suggest that the enhanced iron uptake of splenic HSCs is not specific to PHZ-induced anemia and may represent a general response of HSCs to erythropoietic stress.

### TET2 promotes erythropoiesis by splenic HSC during anemia

Erythropoiesis is associated with global DNA demethylation[51–53]. However, whether DNA demethylation takes place in HSCs upon anemic stress is unknown. Given the critical role of TET2 in regulating DNA demethylation in HSCs, we examined whether TET2 is involved in the responses of HSCs to hemolytic anemia. We deleted *Tet2* from *Mx1-Cre; Tet2fl/fl* mice by three doses of poly (I:C) injections every other day, following by two PHZ injections 2 weeks after the last poly (I:C) injection. Efficient deletion of *Tet2* was confirmed by qPCR and genomic PCR (Supplementary Fig. 4A, B). *Tet2*-deficient mice had slightly increased bone marrow and splenic HSC numbers prior to PHZ treatment, consistent with previous findings[23–25]. Similarly to wild-type mice, we found that splenic HSCs expanded in *Tet2*-deficient mice and showed increased iron content in response to PHZ treatment (Fig. 5a, b). To investigate the role of *Tet2* in augmenting erythroid differentiation of splenic HSC after PHZ treatment, we transplanted splenic HSCs from *Tet2*-deficient and wild-type mice with or without PHZ treatment into irradiated recipient mice. We found that transplantation of splenic HSCs from PHZ-treated mice led to significantly increased reconstitution of multiple erythroid progenitors, such as proerythroblasts (ProE), erythroblasts A, and B (EryA, B) in bone marrow (Fig. 5c). Deleting *Tet2* attenuated this increased erythroid reconstitution by splenic HSCs (Fig. 5c), suggesting that *Tet2* is required to promote erythroid differentiation of splenic HSCs during hemolytic anemia. This *Tet2*-dependent erythroid regenerating potential of splenic HSCs was further confirmed by in vitro assays. The increased ability of splenic HSCs to form erythroid colonies after PHZ treatment was significantly blunted by *Tet2* deletion (Fig. 5d). *Tet2* deletion also reduced the ability of PHZ-treated splenic HSCs to form multipotent colonies containing erythroid cells but increased the number of colonies only containing granulocytes and macrophages/monocytes (Supplementary Fig. 4C–E). *Tet2* deletion did not affect the increased colony forming ability of splenic HSCs upon PHZ treatment (Supplementary Fig. 4F). In addition, PHZ-treated splenic HSCs formed significantly more erythroid progenitors (CFU-E and ProE) than control bone marrow or splenic HSCs when cultured in liquid media containing SCF, IL-6, and EPO in a *Tet2*-dependent manner (Fig. 5e). *Tet2* deletion also reduced the number of stress BFU-E forming cells that respond to sonic hedgehog and BMP[4] (Fig. 5f). Attesting to the importance of *Tet2* in erythroid regeneration, *Tet2* deficiency also impaired the recovery from hemolytic anemia (Fig. 5g). These results establish that TET2 is required for the augmented erythroid differentiation but not expansion of splenic HSCs that occur during hemolytic anemia.

### Increased TET2 protein after anemia promotes erythropoiesis

The dependency of splenic HSCs on *Tet2* to undergo erythroid differentiation during hemolytic anemia prompted us to evaluate levels

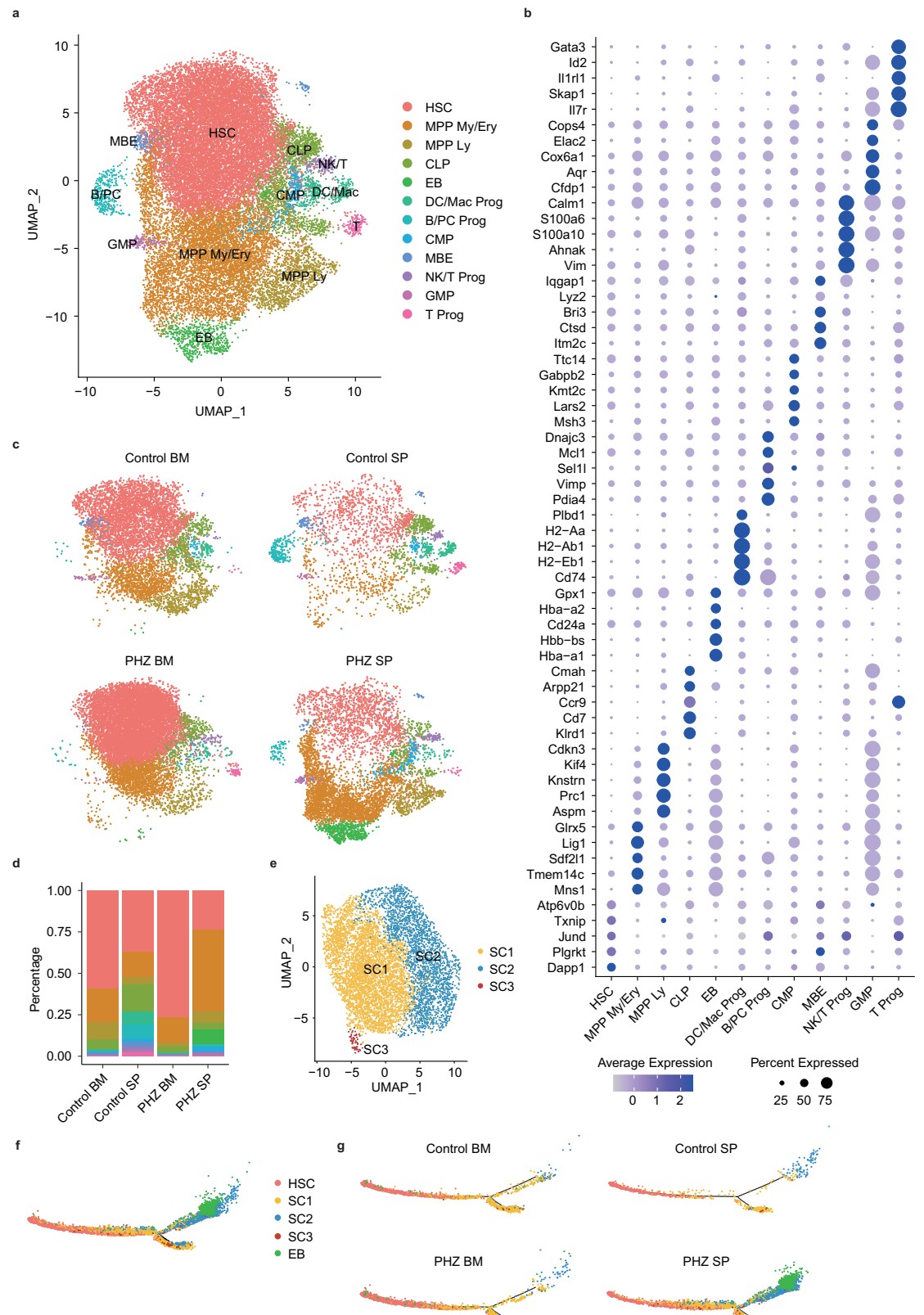

of TET2 expression in these cells. Quantitative PCR analysis of bone marrow and splenic HSCs from PHZ-treated mice did not reveal significant differences in *Tet2* mRNA expression (Fig. 6a). However, we found that splenic HSCs and erythroid progenitors (ProE and CFU-E) had significantly elevated TET2 protein levels after PHZ treatment, whereas protein levels remained unchanged in bone marrow HSCs (Fig. 6b, c and Supplementary Fig. 5A).

TET2 protein is degraded in a manner dependent on calpain and caspase during stem cell differentiation, while it is stabilized by binding to 14-3-3 proteins[54–57]. Consistently, we found that treating LSK cells with a calpain inhibitor in vitro increased TET2 protein (Supplementary Fig. 5B), suggesting that TET2 protein levels may be dynamically regulated post-transcriptionally. The relevance of this mechanism in vivo remains to be investigated.

**Fig. 3 | Single cell analysis of erythroid commitment within bone marrow and splenic HSPCs. a** UMAP embedding all of bone marrow and splenic HSPCs from control or PHZ-treated mice. Cells are labeled on the basis of their cluster identity (*n* = 4). The clusters correspond to hematopoietic stem cell (HSC), myeloerythroid progenitor (MPP-My/Ery), lymphoid-primed multipotent progenitor (MPP Ly), common lymphoid progenitor (CLP), erythroblast (EB), dendritic cell/macrophage progenitor (DC/Mac Prog), B cell/plasma cell progenitor (B/PC Prog), common myeloid progenitor (CMP), mast cell/basophil/eosinophil progenitor (MBE), NK/T cell progenitor (NK/T Prog), granulocyte/macrophage progenitor (GMP), T cell progenitor (T prog) as shown in (**b**). **b** Dot plot showing the average expression of selected marker genes in individual HSPCs. The size of dots represents the percentage of cells expressing a given gene. The cluster-wise mean expression value is shown in log scale and dots are colored on the basis of this value. **c** UMAP embedding HSPCs of each group. **d** Bar graph showing the distribution of transcriptional clusters of each group. **e** UMAP created by sub-clustering of the cluster MPP-My/Ery. Cells are divided into three clusters labeled as sub-cluster (SC) 1, 2, and 3. **f** The cells of clusters HSC, SC1-3, and EB are embedded based on their estimated psuedotime values. **g** Pseudotime trajectory of each group.

We then asked if increased TET2 activity promotes erythroid differentiation of HSCs. To this end, we cultured HSCs with ferrous ammonium sulfate (FAS) together with ascorbate, which reduces Fe(III) to Fe(II) and promotes TET2 activity[34]. Ascorbate has been shown to ameliorate the aberrant self-renewal of *Tet2*-deficient HSCs[36], and we subsequently confirmed that the pronounced self-renewal phenotype exhibited by *Tet2*-deficient HSCs was attenuated by ascorbate treatment (Supplementary Fig. 5C). HSCs treated with FAS plus ascorbate produced significantly more pre-CFU-E cells but not myeloid or CD41+ cells compared to controls. The effect of FAS plus ascorbic acid was not observed in *Tet2*-deficient HSCs (Fig. 6d, e). We then sought to test whether iron stimulates expression of erythropoiesis-related genes in a *Tet2*-dependent manner. Treating HSCs with FAS and ascorbate significantly increased expression of *Slc25a37*, *Tfrc*, *Alas2*, *Ermap*, *Slc4a1*, and *Add2*, in a *Tet2*-dependent manner (Fig. 6f). The expression of those genes was most significantly upregulated when HSCs were treated with the combination of FAS and ascorbate (Supplementary Fig. 5D). These results demonstrate that increased iron uptake in splenic HSCs during anemia promotes TET2 function, inducing expression of erythropoiesis- and iron metabolism-related genes in HSCs.

### TET2 mediates erythroid gene expression in splenic HSCs
To gain further insight into the mechanism by which hemolytic anemia augments the regenerative and erythroid potential of splenic HSC, we performed RNA-seq on PHZ-treated bone marrow and splenic HSCs and bone marrow HSCs from untreated mice (Fig. 7a, b). Untreated splenic HSCs were not included in the analysis due to scarcity of the cells. Gene set enrichment analysis (GSEA) revealed that genes involved in heme metabolism were enriched in PHZ-treated splenic HSCs compared to untreated bone marrow HSCs (Fig. 7c). Gene Ontology (GO) analysis also detected gene signatures associated with iron transport in PHZ-treated splenic HSCs (Fig. 7d). Erythroid-related and iron metabolism-related genes including *Tfrc*, *Alas2*, *Slc25a37*, *Ermap*, *Slc4a1*, *Car1*, *Add2* and *Cldn13* were differentially expressed in PHZ-treated splenic HSCs compared to control bone marrow HSCs (Fig. 7a, b). A subset of Gata1 regulated genes in a Gata1-inducible erythroblast cell line[58] were differentially expressed in PHZ-treated splenic HSCs compared to bone marrow HSCs (Supplementary Fig. 5E). Nine of the top 30 most differentially upregulated genes exhibited erythroid-biased expression (Supplementary Fig. 6A–I). Quantitative PCR analysis to confirm transcriptional changes in PHZ-treated splenic HSCs showed strong induction of erythropoiesis- and iron-related genes such as *Slc4a1*, *Cldn13*, *Ermap*, and *Tfrc* (Fig. 7e). Consistent with the increased expression of erythroid genes, motif analysis identified enrichment of recognition sites of MAFK[59] and EKLF in PHZ-treated splenic HSCs (Fig. 7f).

To test whether the induction of erythropoiesis-related and iron-related genes depends on *Tet2*, we examined the expression of these genes in HSCs isolated from PHZ-treated *Tet2*-deficient mice. Loss of *Tet2* significantly attenuated the induction of 5 out of 7 PHZ-induced erythropoiesis- and iron-related genes, including *Slc4a1* (Erythrocyte Membrane Protein Band 3), *Cldn13*, and *Tfrc* (Fig. 7e and Supplementary Fig. 6J). Consistently, cell surface expression of CD71 (encoded by

*Tfrc*) was significantly reduced in *Tet2*-deficient splenic HSCs after PHZ treatment (Fig. 7g). Finally, we tested whether TET2 induces the expression of erythroid genes in splenic HSCs after PHZ treatment by DNA demethylation. Analysis of a 5hmC profiling dataset in LSK and MEP cells[60] identified a robust 5hmC peak, particularly in MEPs, in the promoter region of *Tfrc* that contained 5 CpG sites (Fig. 7h). We then performed bisulfite sequencing of this locus in wild-type and *Tet2*-deficient splenic HSCs with or without PHZ treatment. This analysis revealed that while all of the CpG sites in this locus were highly methylated in wild-type splenic HSCs, one of the CpG site (Chr16: 32,608,140) became significantly demethylated after PHZ treatment (Fig. 7i, j and Supplementary Fig. 6K). Demethylation of this CpG site was abolished in PHZ-treated *Tet2*-deficient splenic HSCs, demonstrating the TET2 dependency of DNA demethylation during hemolytic anemia (Fig. 7i, j). Collectively, these results support a model in which increased iron uptake and TET2 expression in splenic HSCs during anemia promote DNA demethylation and enhanced erythropoiesis.

## Discussion
In this study, we demonstrate that splenic HSCs are expanded and activated to respond to stress erythropoiesis during hemolytic anemia. In splenic HSCs, increased iron uptake and TET2 protein expression coordinate to promote erythroid differentiation via enhancing DNA demethylation on iron metabolism and erythroid-related genes. We found that blocking iron uptake reduces PHZ-induced splenic HSC expansion, activation and erythroid-biased differentiation, whereas *Tet2* knockout mainly eliminates their enhanced erythroid potential. Our findings demonstrate the active role played by HSCs in regenerating the erythroid compartment during stress erythropoiesis and show that this activity is regulated by iron and TET2.

Prior studies focusing on the role of erythroid progenitors during stress erythropoiesis identified immature self-renewing erythroid progenitors as responsible for rapid erythropoiesis under stress conditions[4,5,9]. These progenitors are derived from MPPs that migrate from bone marrow to spleen and are distinct from steady-state bone marrow erythroid progenitors[5,9]. While our study mainly focused on how HSCs respond to acute hemolytic anemia, consistent with prior studies, we also observed an increase of MPP and HPC2, a myeloerythroid-biased progenitor population, in the spleen upon PHZ treatment. These results demonstrate that the effects of erythroid stress manifests beyond MPPs by directing HSCs to regenerate the erythroid compartment. We postulate that anemia rapidly activates MPPs and erythroid progenitors to provide the first line of defense against anemia, and that HSCs are also activated to provide a longer-term erythroid output for full recovery.

A variety of hematopoietic stresses including cancer, chemotherapies, infection, anemia, myocardial infarction, and pregnancy causes extramedullary hematopoiesis (EMH) in the spleen or liver in mice[4,9–11,61–64]. Similarly, EMH in humans is well documented in cases of myeloproliferative neoplasm (MPN) or other neoplasms[65,66]. Although it is generally considered that the bone marrow increases erythroid output upon anemia in humans, as evidenced by bone marrow hyperplasia[67], examples of EMH in response to anemia exists in humans. For example, EMH in thalassemia, sickle cell anemia, and

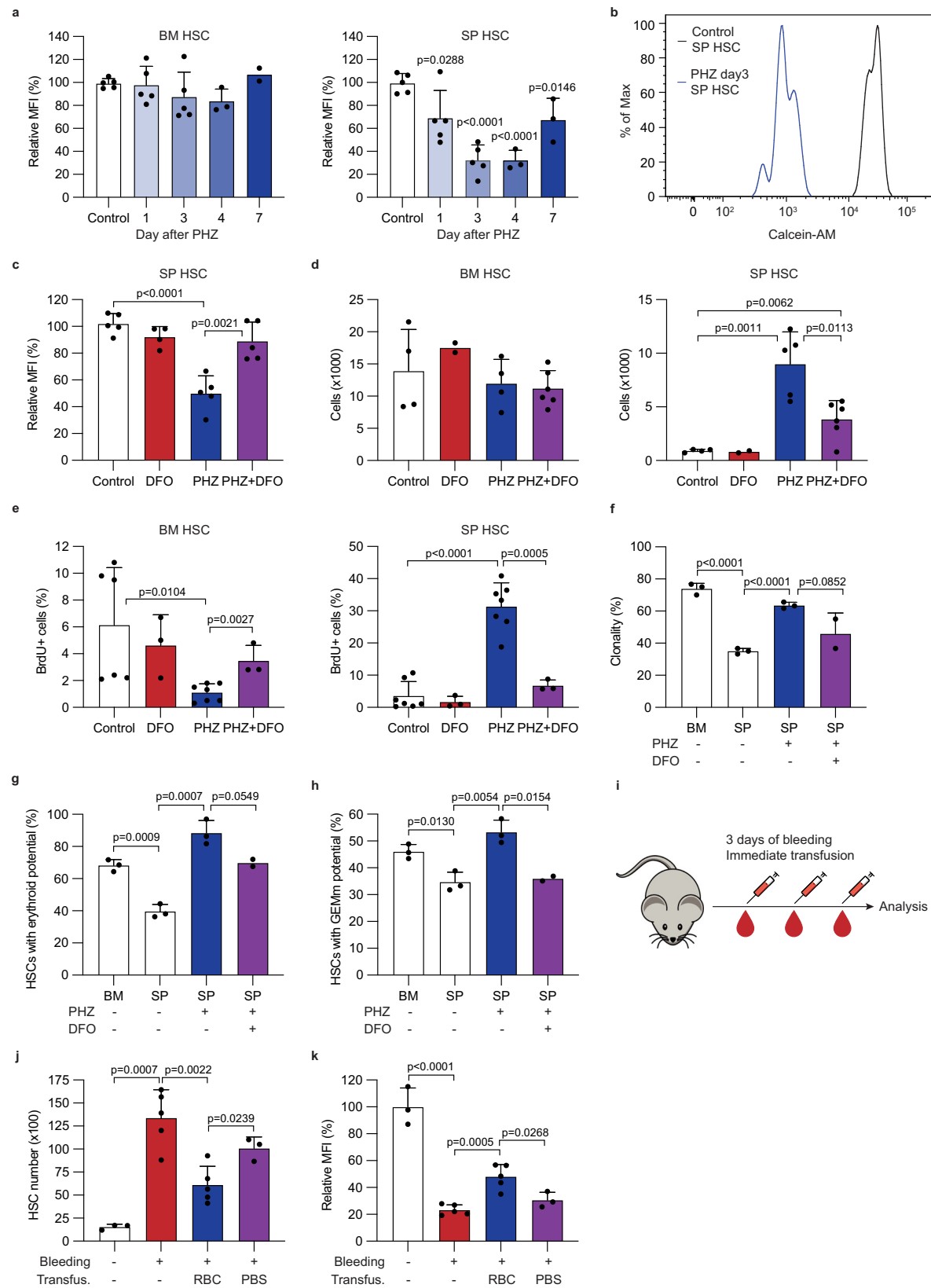

pernicious anemia have been long recognized[66,68–72]. The most noted EMH sites in humans include the spleen, liver, lymph nodes, and the paraspinal region[66,73]. A recent study evaluated 1933 cases of extramedullary hematopoiesis, among which 309 cases were without MPN and 8%, 7%, and 3% of those cases were associated with hemolytic anemia, thalassemia, or hereditary spherocytosis, all of which are

regarded as hemolytic disorders[65]. Importantly, the spleen was the most frequent sites of extramedullary hematopoiesis in this study (92% of hemolytic anemia- and 50% of thalassemia-associated extramedullary hematopoiesis cases involved the spleen), consistent with the observation in mice[65]. Additionally, a recent study using single cell transcriptomics and colony forming assays demonstrated that human

**Fig. 4 | Increased iron uptake is responsible for the functional changes of splenic HSC. a** Relative changes of the mean fluorescence intensity (MFI) of calcein AM staining in bone marrow (left) and spleen (right) HSCs over time ($n = 5$ for control and days 1 and 3 in both groups, $n = 3$ for day 4 in both groups and for day 7 of spleen, and $n = 2$ for day 7 in bone marrow). **b** Representative histograms of calcein AM staining of control or PHZ-treated (day 3 post treatment) splenic HSCs. **c** Relative changes of calcein AM MFI in splenic HSCs after DFO, PHZ, or PHZ + DFO treatments ($n = 4$ for PHZ and $n = 5$ for other groups). **d** HSC numbers in the bone marrow (left) and spleen (right) after DFO, PHZ, or PHZ + DFO treatments ($n = 4, 2,$ 4, and 6 for control, DFO, PHZ, and PHZ + DFO, respectively in bone marrow; $n = 4,$ 2, 5, and 6 for control, DFO, PHZ, and PHZ + DFO, respectively in spleen). **e** BrdU incorporation into bone marrow (left) or splenic (right) HSCs after DFO, PHZ, or PHZ + DFO treatments ($n = 6, 3, 7,$ and 3 for control, DFO, PHZ, and PHZ + DFO, respectively in bone marrow; n = 7, 3, 7, and 3 for control, DFO, PHZ, and PHZ + DFO, respectively in spleen). **f**–**h** Colony forming assays with single-cell sorted bone

marrow or splenic HSCs from control, PHZ-treated, or PHZ + DFO-treated mice. Shown are clonality (**f**), frequency of colonies containing erythroid cells (**g**), and frequency of GEMm colonies (**h**) ($n = 2$ for PHZ + DFO-treated group and $n = 3$ for others). **i** A schematic of the serial bleeding and transfusion analysis. Mice were bled for 3 days followed by immediate transfusion of fresh RBCs or PBS. **j, k** Splenic HSC numbers (**j**) and calcein AM MFI (**k**) after repeated bleeding and transfusion of RBCs or PBS ($n = 3, 5, 3,$ and 5 for control, bleeding, bleeding and RBC transfusion, and bleeding and PBS transfusion, respectively). Comparisons were performed by unpaired, two-tailed Student's $t$ test. All data represent mean ± standard deviation. All numbers ($n$) are independent animals. Source data are provided as a Source Data file. Mouse and syringe illustrations were adapted from Creazilla (https://creazilla.com/nodes/18581-big-eared-grey-mouse-clipart, https://creazilla.com/nodes/42124-syringe-emoji-clipart) under the Creative Commons license CC0 and CC BY, respectively.

splenic HSPCs in patients with hereditary spherocytosis displaying splenomegaly due to chronic anemia exhibit increased erythroid colony forming capacity and erythroid gene priming at a single cell level[74]. Thus, the ability to respond to anemia and become primed to the erythroid lineage appears to be a conserved feature of splenic HSCs. We note, however, that it remains unclear to what extent extramedullary erythropoiesis contributes to overall erythroid output in the diseased state.

We provide evidence that iron has an instructive role in promoting erythropoiesis by HSCs. Elevated splenic iron levels upon acute hemolytic anemia, due to erythrophagocytosis in the spleen and liver[49,75,76], enhanced iron uptake and proliferation of splenic HSCs. This increase is likely mediated by upregulation of the transferrin receptor CD71 (*Tfrc*), as we found its cell surface and mRNA expression to increase in splenic HSCs after PHZ treatment. Indeed, CD71-mediated iron uptake is crucial for HSC function[20] and we show that iron chelation suppresses the elevated proliferation and regenerative potential of HSCs after hemolytic anemia. Of note, increases in iron levels in HSCs are transient and return to baseline as mice recover from anemia. On the contrary, chronic systemic iron overload observed in diseases such as β-thalassemia or myelodysplastic syndromes is often linked to impaired HSC function[17,77,78]. Chronic iron overload may constitutively stimulate HSCs, leading to their exhaustion and providing an explanation as to why pathological conditions with iron overload impair HSC function.

We show that TET2 protein level is increased in splenic HSCs and that *Tet2* is required to promote the erythroid potential of HSCs under stress erythropoiesis. While we show that calpains might be involved in destabilizing TET2 protein in HSPCs, whether the inhibition of this mechanism is responsible for TET2 stabilization in HSCs during anemia remains to be seen. Additionally, whether the TET2-dependent mechanisms we present here occurs in patients with *TET2* mutations, such as those with myelodysplastic syndromes or clonal hematopoiesis, remain unknown. Nonetheless, recent studies have established that ascorbate can augment TET2 activity in vivo. Administration of ascorbate activates TET activity and suppresses aberrant HSC self-renewal and myeloid neoplasm in *Tet2*[+/−] mice[35,36]. Combinatory treatment of iron and ascorbate also improves anemia in rodents and humans[79–81]. Future studies are needed to establish the therapeutic potential of TET2 activation or stabilization to stimulate erythropoiesis.

## Methods
### Animals
Mice were housed in AAALAC-accredited, specific-pathogen-free animal care facilities at Baylor College of Medicine (BCM) with 12-h light–dark cycle, ambient temperature at 70 °F, humidity within 30–80%, and received standard chow ad libitum. All procedures were approved by the BCM (protocol #AN-5858) Institutional Animal Care and Use Committees. The mouse alleles used in this study were *UBC-*

*GFP* (C57BL/6-Tg(UBC-GFP)30Scha/J, JAX stock 004353), *Krt18-CreER* (Tg(KRT18-cre/ERT2)23Blpn/J, JAX stock 017948), *Rosa26-LSL-tdTomato* (JAX stock 007909), *Tet2*[fl/fl] (B6;129S-Tet2[tm1.1Iaai]/J, JAX stock 017573), and *Mx1-Cre* (B6.Cg-Tg(Mx1-cre)1Cgn/J, JAX stock 003556) on a C57BL/6 background. CD45.1 (B6.SJL-Ptprca Pepcb/BoyJ, JAX stock 002014) and BL6 mice (C57BL/6J, JAX stock 000664) were used as transplant recipient mice. Mice 8 to 12 weeks of age of both sexes were used, and experimental mice were separated by sex and housed with up to five mice per cage.

### Drug and chemical treatments
The *Mx1-Cre; Tet2*[fl/fl] mice were treated with poly(I:C) resuspended in PBS at 100 µg/ml by intraperitoneal injection with 20 µg poly(I:C) every other day for three doses to induce *Tet2* deletion. In total, 60 mg/kg of PHZ was injected intraperitoneally at 2 weeks after the last poly(I:C) injection for two constitutive days. PHZ-treated mice were analyzed at days 1–7 post treatment. For PHZ and DFO co-treatment, DFO was injected intraperitoneally at 100 mg/kg daily from the day of the first PHZ injection until the day of analysis. Krt18-tdT mice were injected with tamoxifen in corn oil at 50 mg/kg by intraperitoneal injection for 5 constitutive days to induce HSC labeling, and analyzed at day 5 and day 14 after the first dose of tamoxifen.

### Transplantation and limiting dilution transplantation
In all, 8–16-week-old C57BL/6 CD45.1 mice were randomly assigned to groups after receiving two doses of radiation (500 cGy) administered at least 2 h apart (total 1000 cGy). For limiting dilution transplantation, different doses (bone marrow cells: $7.5 \times 10^3$, $1.5 \times 10^4$, $3 \times 10^4$, $1 \times 10^5$, $2 \times 10^5$; splenocytes: $3 \times 10^5$, $6 \times 10^5$, $1.2 \times 10^6$, and $3 \times 10^6$) of bone marrow cells or splenocytes were transplanted to lethally irradiated recipients along with $2 \times 10^5$ competitor cells and analyzed at 4 months after transplantation. For HSCs transplantation, 50 bone marrow or splenic HSCs sorted from UBC-GFP mice treated or untreated with PHZ were transplanted along with $2 \times 10^5$ competitor cells. In all, 50 HSCs from *Tet2*-deficient mice were transplanted in the same manner.

### Flow cytometry and HSC isolation
Bone marrow cells were either flushed from the long bones (tibias and femurs) or isolated by crushing the long bones (tibias and femurs), pelvic bones, vertebrae, and sternum with mortar and pestle in Hank's buffered salt solution (HBSS) without calcium and magnesium, supplemented with 2% heat-inactivated bovine serum (GIBCO, Grand Island, NY). Splenocytes were dissociated by smashing the entire spleen between slides. Cells were triturated and filtered through a nylon screen (100 µm, Sefar America, Kansas City, MO) or a 40-µm cell strainer (Fisher Scientific, Pittsburg, PA) to obtain a single-cell suspension. For isolation of lineage⁻Sca-1⁺c-kit⁺(LSK) CD150⁺CD48⁻/low HSCs, bone marrow cells were incubated with PE-Cy5-conjugated anti-

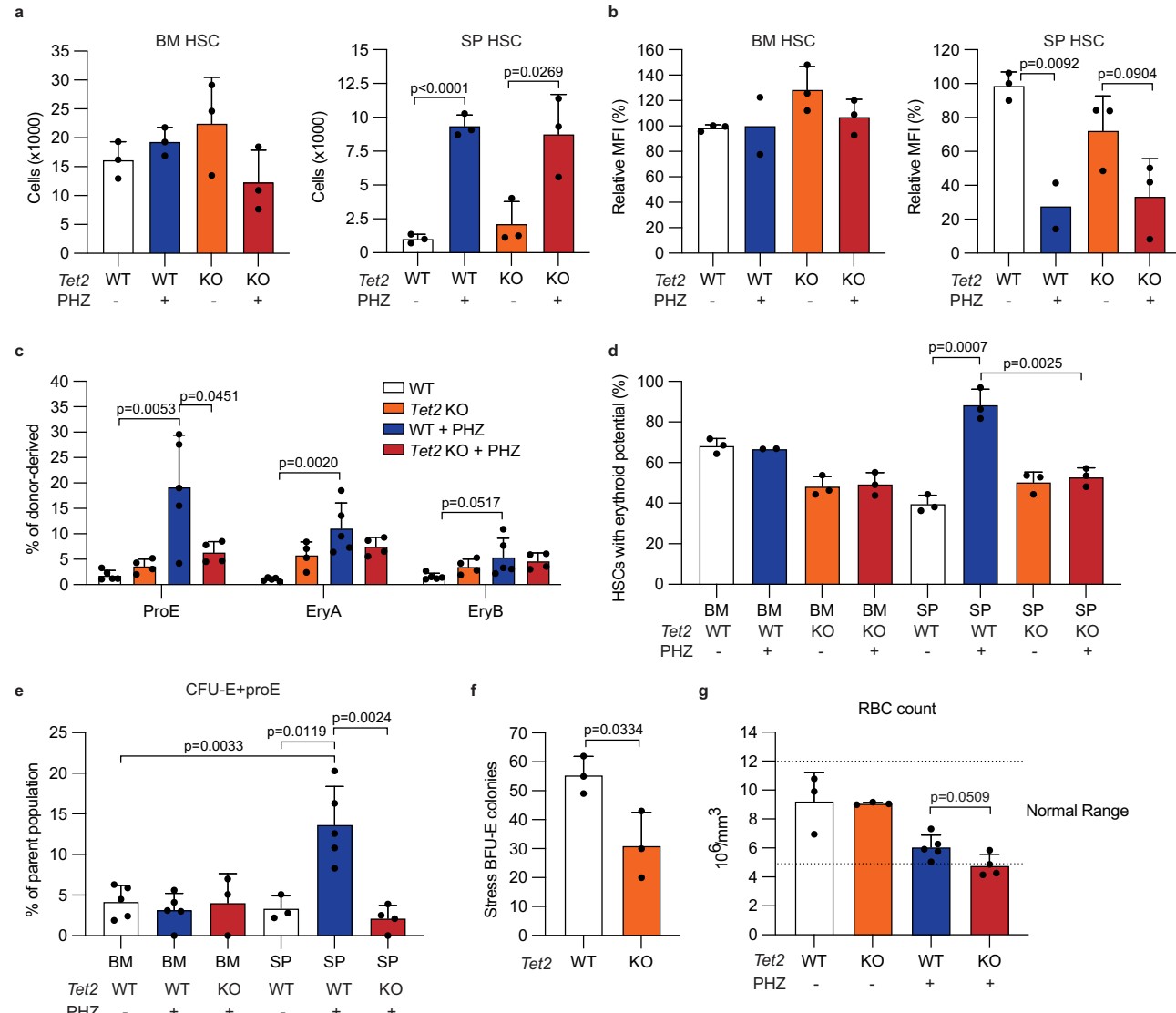

**Fig. 5 | Increased erythroid potential of splenic HSC during anemia depends on TET2. a, b** The numbers of HSCs (**a**) and relative changes of calcein AM MFI in HSCs (**b**) from bone marrow (left) or spleen (right) of wild-type or *Tet2* knockout mice treated or untreated with PHZ ($n = 2$ for PHZ-treated wild-type in (**b**) and $n = 3$ for other groups in (**a**, **b**). **c** Transplantation of 50 splenic HSCs isolated from wild-type or *Tet2* knockout mice treated or untreated with PHZ. Reconstitution level of erythroid progenitors were analyzed at 2 weeks post transplantation ($n = 5$ for wild-type and $n = 4$ for *Tet2* knockout). The quantification of erythroid progenitor reconstitution levels is shown. **d** Colony forming assays with single-cell sorted bone marrow or splenic HSCs of wild-type or *Tet2* knockout mice treated or untreated with PHZ. Frequency of colonies containing erythroid cells was shown ($n = 2$ for PHZ-treated wild-type bone marrow and $n = 3$ for other groups). **e** The frequency of erythroid progenitor cells (lin⁻Sca-1⁻c-kit⁺CD41⁻CD16/32⁻CD150⁻CD105⁺ cells [CFU-E +proE]) formed in vitro after 14 days culture of wild-type or *Tet2* knockout bone marrow or splenic HSCs with or without PHZ treatment ($n = 5, 5, 3, 3, 5,$ and 4 for untreated wild-type bone marrow, PHZ-treated wild-type bone marrow, PHZ-treated *Tet2* knockout bone marrow, untreated wild-type spleen, PHZ-treated wild-type spleen, and PHZ-treated *Tet2* knockout spleen, respectively). **f** The numbers of BMP4- and sonic hedgehog-responsive stress BFU-E colonies from wild-type or *Tet2* knockout cells ($n = 3$). **g** Red blood cell count of control or PHZ-treated wild-type or *Tet2* knockout mice 5 days after PHZ treatment ($n = 3, 3, 5,$ and 4 for untreated wild-type, untreated *Tet2* knockout, PHZ-treated wild-type, and PHZ-treated *Tet2* knockout, respectively). The dotted line indicates the range of red blood cells considered normal. Comparisons were performed by unpaired, two-tailed Student's *t* test. All data represent mean ± standard deviation. All numbers (*n*) are independent animals. Source data are provided as a Source Data file.

CD150 (TC15-12F12.2; BioLegend, San Diego, CA), PE-Cy7-conjugated anti-CD48, APC-conjugated anti-Sca-1 (Ly6A/E; E13-6.7), and biotin-conjugated anti-c-kit (2B8) antibody, in addition to antibodies against the following FITC-conjugated lineage markers: Ter119, B220 (6B2), Gr-1 (8C5), CD2 (RM2-5), CD3 (KT31.1), and CD8 (53-6.7). Biotin-conjugated antibodies were visualized using streptavidin-conjugated APC-Cy7. For HSC sorting, bone marrow cells were pre-enriched by selecting c-kit⁺ cells using paramagnetic microbeads and autoMACS (Miltenyi Biotec, Auburn, CA). Nonviable cells were excluded from sorts and analyses using the viability dye 4′,6-diamidino-2-phenylindole (DAPI) (1 μg/ml).

For iron level analysis, cells were pre-incubated with 0.2 μM calcein AM (BioLegend) at 37 °C for 15 min before staining for HSC markers. For hematopoietic lineage tracing, 17 populations were analyzed along with HSCs: MPPs (LSKCD150⁻CD48⁻/low), HPC2 (LSKCD150⁺CD48⁺), HPC1 (LSKCD150⁻CD48⁺), GMP (Lin⁻Sca1⁻c-kit⁺CD34⁺CD16/32⁺), CMP (Lin⁻Sca1⁻c-kit⁺CD34⁺CD16/32⁻/low), MEP (Lin⁻Sca1⁻c-kit⁺CD34⁻/low CD16/32⁻/low), CLP (Lin⁻Sca1low c-kitlow CD127⁺CD135⁺), MkP (Lin⁻Sca1⁻c-kit⁺ CD150⁺CD41⁺), pre CFU-E (Lin⁻Sca1⁻c-kit⁺CD41⁻CD16/32⁻CD150⁺CD105⁺), pre MegE (Lin⁻Sca1⁻c-kit⁺CD41⁻CD16/32⁻CD150⁺CD105⁻/low), pre GM (Lin⁻Sca1⁻c-kit⁺CD41⁻CD16/32⁻CD150⁻CD105⁻), ProE (CD71⁺Ter119medium)[82],

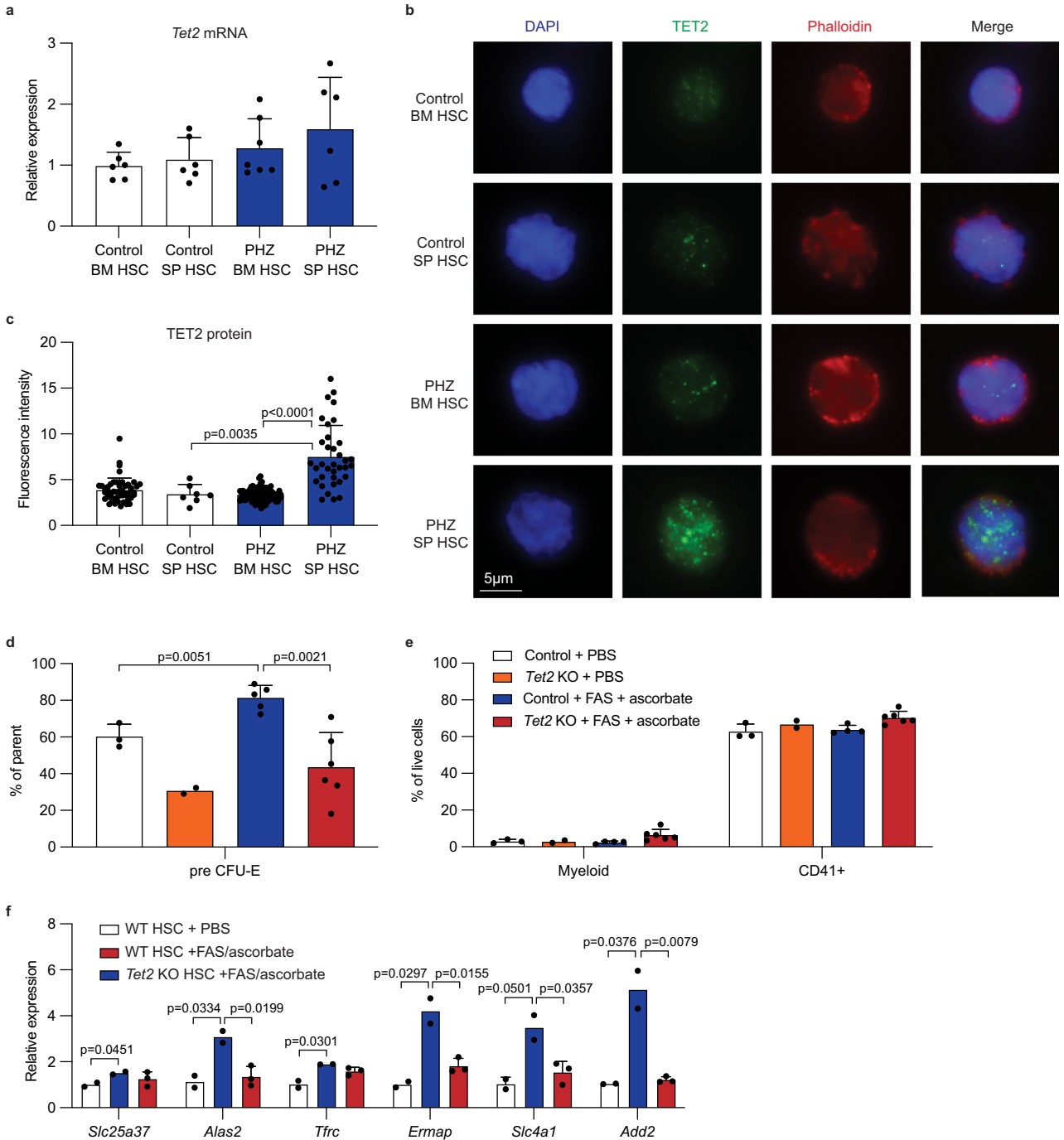

**Fig. 6 | Increased TET2 protein expression and activity promote erythropoiesis during hemolytic anemia. a** Relative expression of *Tet2* mRNA in bone marrow or splenic HSCs with or without PHZ treatment. Data is presented as relative $2^{-\Delta\Delta Ct}$ compared to control bone marrow HSC ($n = 7$ for PHZ-treated bone marrow HSC, $n = 6$ for other groups). **b, c** Representative immunofluorescence images (**b**) and quantification (**c**) of TET2 protein in bone marrow or splenic HSCs with or without PHZ treatment ($n = 49, 7, 83,$ and 35 cells for untreated bone marrow HSCs, untreated splenic HSCs, PHZ-treated bone marrow HSCs, PHZ-treated splenic HSCs, respectively). **d, e** Wild-type or *Tet2* knockout HSCs were cultured in media containing PBS or FAS plus ascorbate, together with SCF and TPO. Frequencies of pre CFU-E (**d**) and mature myeloid and CD41 positive cells (**e**) are shown ($n = 3, 2,$ 5, and 6 for wild-type +PBS, *Tet2* knockout +PBS, wild-type +FAS/ascorbate, and *Tet2* knockout +FAS/ascorbate, respectively in (**d**); $n = 3, 2, 4,$ and 6 for wild-type +PBS, *Tet2* knockout +PBS, wild-type +FAS/ascorbate, and *Tet2* knockout +FAS/ascorbate, respectively in (**e**). **f** Relative expression of genes involved in erythropoiesis or iron metabolism in HSCs cultured in FAS plus ascorbate. Data is presented as relative $2^{-\Delta\Delta Ct}$ compared to control PBS ($n = 2, 2,$ and 3 for wild-type +PBS, wild-type +FAS/ascorbate, and *Tet2* knockout +FAS/ascorbate, respectively). Comparisons were performed by unpaired, two-tailed Student's *t* test. All data represent mean ± standard deviation. Numbers (*n*) are independent animals in (**a, d, e, f**) and cells from one animal per group in (**b**). Source data are provided as a Source Data file.

EryA (Ter119$^{high}$CD71$^{+}$FSC$^{high}$), EryB (Ter119$^{high}$CD71$^{+}$FSC$^{low}$), EryC (Ter119$^{high}$CD71$^{-/low}$FSC$^{low}$)[83], and lineage+ cells (CD3+, B220+, Mac1/Gr1+, CD41+, and Ter119+), CFU-E (Lin$^{-}$Sca1$^{-}$c-kit$^{+}$CD41$^{-}$CD16/32$^{-}$CD150$^{-/low}$CD105$^{+}$CD71$^{+}$Ter119$^{-/low}$)[84], stress erythroid progenitor population I (c-kit$^{+}$CD71$^{low/med}$Ter119$^{low/-}$), population II (c-kit$^{+}$CD71$^{hi}$Ter119$^{med}$), and population III (c-kit$^{+}$CD71$^{low/med}$Ter119$^{high}$). Unless otherwise noted, antibodies were obtained from BioLegend, BD Biosciences, or eBioscience (San Diego, CA). Flow cytometry was performed with

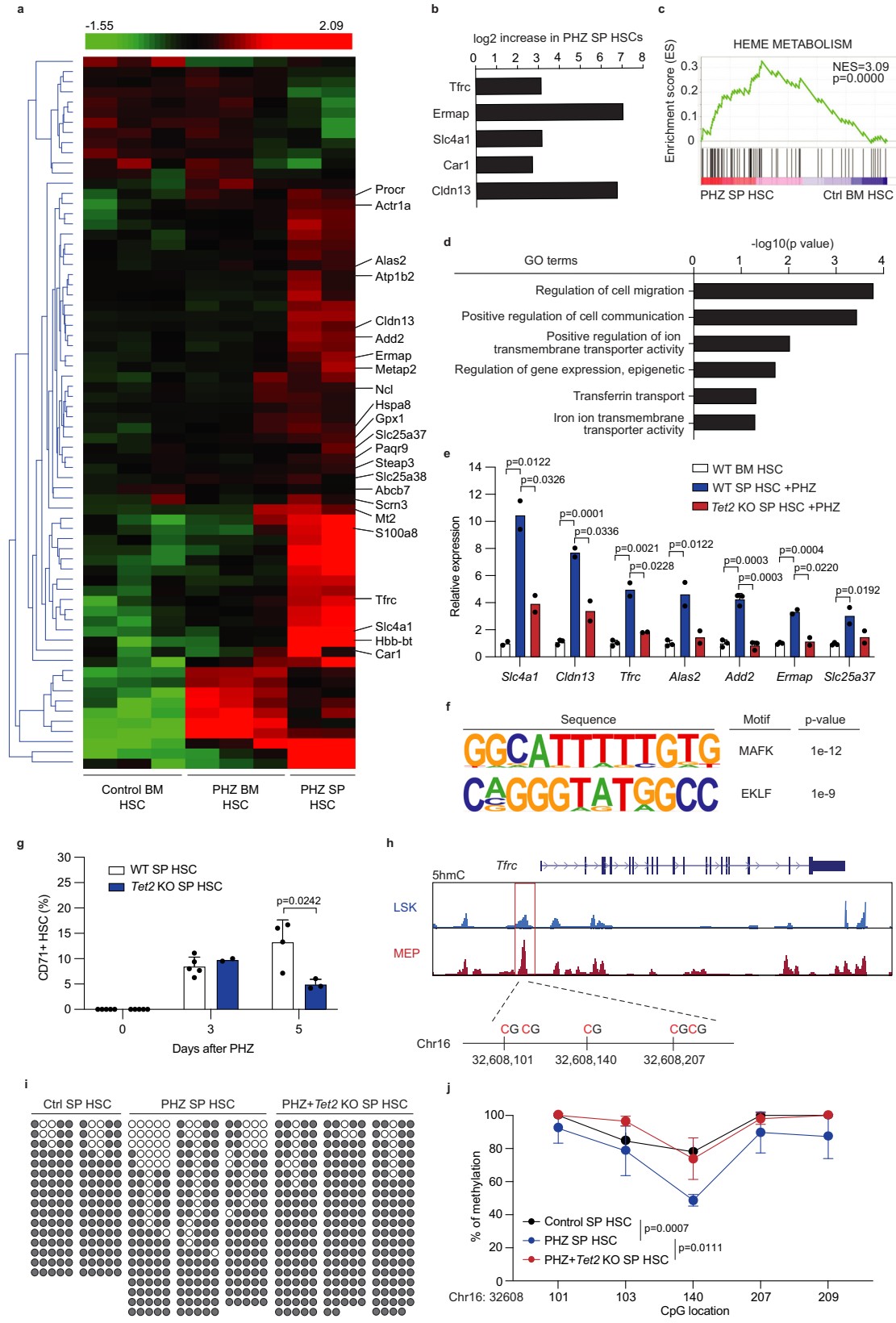

FACSAria II, FACSCanto II, LSR II, or LSRFortessa flow-cytometers (BD Biosciences).

### BrdU analysis

In total, 2 mg of BrdU (in PBS) were injected intraperitoneal to control or PHZ-treated mice 2 days after the second dose of PHZ and maintained on 1 mg/ml BrdU in the drinking water. Bone marrow and spleen were collected 24 h post BrdU injection and stained for HSC, MPP, and HPC1/2. HSCs were then pre-enriched by selecting c-kit+ cells using paramagnetic microbeads on autoMACS then fixed and permeabilized using the APC BrdU flow kit (BD Biosciences). Fixed cells were treated with DNase I, stained with anti-BrdU APC (BD

**Fig. 7 | TET2-dependent induction and DNA demethylation of erythroid genes in splenic HSCs during hemolytic anemia. a** Heatmap for the differentially expressed genes in control bone marrow HSCs, PHZ-treated bone marrow HSCs, and PHZ-treated splenic HSCs. Genes involved in erythropoiesis or iron metabolism are labeled. **b** Log$_2$ fold change (L2FC) of the differentially expressed erythroid genes in PHZ-treated splenic HSCs. **c** GSEA showing the enrichment of genes involved in heme metabolism in PHZ-treated splenic HSCs. **d** Selected GO terms that were upregulated in PHZ-treated splenic HSCs. **e** Relative expression of genes involved in erythropoiesis or iron metabolism in bone marrow or splenic HSCs with or without PHZ treatment. Data is presented as relative $2^{-\Delta\Delta Ct}$ compared to control bone marrow ($n = 2$ for *Slc4a1* and $n = 3$ for other genes in wild-type bone marrow; $n = 3$ for *Add2* and $n = 2$ for other genes in PHZ-treated wild-type spleen and PHZ-treated *Tet2* knockout spleen). **f** Enrichment of transcription factor motifs involved in erythropoiesis in PHZ-treated splenic HSCs. **g** The frequency of CD71+ splenic HSCs in WT or *Tet2* KO spleens at day 3 or 5 post PHZ treatment ($n = 5, 5,$ and 4 for day 0, 3, and 5 in wild-type spleen, respectively; $n = 5, 2,$ and 3 for day 0, 3, and 5 in *Tet2* knockout spleen, respectively). **h** 5hmC profiles at the *Tfrc* locus in LSK or MEP cells. The chromosomal coordinates of CpG sites within the *Tfrc* regulatory element (red box) is also shown. **i, j** Bisulfite sequencing results of *Tfrc* methylation level at the indicated CpG sites in control or *Tet2* KO splenic HSCs with or without PHZ treatment. Unfilled and filled circles represent unmethylated and methylated CpGs, respectively (**i**). Quantification of the methylation level at each CpG sites are shown (**j**) ($n = 2, 4,$ and 3 for control spleen, PHZ-treated spleen, and PHZ-treated *Tet2* knockout spleen, respectively). The *P* values shown are for the methylation level at Chr16: 32,608,140. Comparisons were performed by unpaired, two-tailed Student's *t* test. All data represent mean ± standard deviation. All numbers (*n*) are independent animals. Source data are provided as a Source Data file.

Biosciences), washed, and stained with DAPI before flow cytometry analysis.

## HSC culture and CFU assay

For CFU assays, HSCs were single-cell sorted directly into 96-well plates containing M3434 MethoCult methylcellulose media (StemCell Technologies, Cambridge, MA) and cultured for 10–14 days before scoring based on morphology. Colonies were collected and washed and stained for flow cytometry or genomic DNA extracted for genotyping. To examine the effects of iron on HSCs in culture, 200 wild-type or *Tet2* KO HSCs were sorted into SF-O3 medium (Iwai North America Inc., San Carlos, CA) containing 1% heat-inactivated fetal bovine serum (GIBCO), 1% penicillin–streptomycin (GIBCO), 100 ng/ml SCF, 20 ng/ml TPO (both from PeproTech), 50 μg/ml ascorbic acid and 5 μM FAS or equivalent volume of PBS. HSCs were cultured for 2 weeks and analyzed by flow cytometry or by qPCR. For assessing the erythroid potential, control or PHZ-treated wild-type or *Tet2* KO HSCs were sorted into SF-O3 medium containing containing 1% heat-inactivated fetal bovine serum (GIBCO), 1% penicillin–streptomycin (GIBCO), 100 ng/ml SCF, 9 ng/ml EPO (BioLegend, CA) and 1.5 ng/ml of IL-6 (PeproTech).

## HSC immunofluorescent staining

HSCs (CD150$^+$CD48$^{-/low}$ LSK) isolated from wild-type or *Tet2* KO mice with or without PHZ treatment were sorted directly on glass slides with a FACSAria II sorter. Most cells dried immediately, allowing their cellular morphology to be preserved. Cells were fixed in ice-cold methanol for 5 min, washed in PBS, and blocked in blocking buffer (4% goat serum, 0.4% BSA, and 0.1% NP-40 in PBS) before being incubated with anti-TET2 antibody (D6C7K, Cell Signaling Technology) diluted 1:500 in blocking buffer overnight at 4 °C. Slides were rinsed and incubated with Alexa Fluor 488-conjugated goat anti-rabbit antibody (Invitrogen, Eugene, OR) at 1:500 dilution in blocking buffer for 1 h. After washing in PBS, slides were incubated with blocking buffer containing DAPI and 1:200 DyLight™ 554 Phalloidin (Cell Signaling Technology) for 20 min at room temperature. Images were obtained using a Leica DMI 6000B microscope. Fluorescence intensity of individual cells was quantified using the ImageJ software.

## Quantitative real-time (reverse transcription) PCR

HSCs and other hematopoietic cells were sorted into Trizol (Life Technologies, Carlsbad, CA) and RNA was isolated according to the manufacturer's instructions. cDNA was made with random primers and SuperScript VILO cDNA Synthesis Kit (ThermoFisher Scientific, Waltham, MA). Quantitative PCR was performed using SYBR Premium Ex Taq (Tli RNaseH Plus) ROX Plus kits (Clontech) on a ViiA7 Real-Time PCR System (ThermoFisher Scientific, Waltham, MA). Each sample was normalized to β-actin. Data were analyzed using the $2^{-\Delta\Delta Ct}$ method. Primers to quantify cDNA levels are listed in Supplementary Table 1.

## Cycloheximide chase assay and immunoblotting

LSK cells from the bone marrow of wild-type mice were sorted into SF-O3 medium with supplementation described above and cultured for 7 days. Cells were then treated with 40 μM calpeptin (calpain inhibitor), 40 μM Z-DEVD-FMK (caspase-3 inhibitor), or equivalent volume of DMSO along with 300 μg/ml of cycloheximide for 0, 4, and 6 h. After incubation, the same number of cells from each test population were collected into 10% trichloroacetic acid (TCA). Extracts were incubated on ice for 30 min and spun down for 10 min at 15,000×*g* at 4 °C. The resulting pellets were washed twice with acetone, dried, and solubilized with a solubilization buffer (9 M Urea, 2% Triton X-100, 1% DTT) before adding an LDS loading buffer (Invitrogen, Carlsbad, CA). Proteins were separated on bis-tris polyacrylamide gel and transferred to a polyvinylidene difluoride membrane (Millipore, Billerica, MA). Antibodies used for immunoblotting are rabbit anti-TET2 (no. 36449, Cell Signaling Technology, 1:1000) and mouse anti-β-actin (A1978, Sigma-Aldrich, 1:1000).

## Stressed red blood cell transfusion

Red blood cell (RBC) transfusion was performed as described[50]. RBCs were purified from wild-type mice using Lymphoprep (StemCell Technologies). Purified RBCs were then heated at 48 °C for 20 min under continuous shaking. The volume of RBCs was adjusted to 17.0–17.5 g/dl of hemoglobin and 400 μl of RBCs was injected intravenously. Bone marrow and spleen from transfused mice were analyzed at 2 h post injection.

## Serial bleeding

Serial bleeding was performed as previously described[12]. In short, mice were placed under a warming lamp and bled for at least 300 μl of blood via tail vein (day 0). Mice were subsequently bled on day 3, 6 and 9 and analyzed on day 10. For RBC transfusion, RBCs were purified from wild-type mice using Lymphoprep (StemCell Technologies) and resuspended in PBS. RBCs from 300 μl of blood were injected intravenously immediately after each bleeding.

## Stress BFU-E assay

Stress BFU-E assays were performed as previously described[6]. In brief, nucleated bone marrow cells from wild-type or *Tet2* knockout mice were cultured in media containing 20% fetal bovine serum, sonic hedgehog (25 ng/mL), BMP4 (15 ng/mL), GDF15 (30 ng/mL), SCF (15 ng/mL), and Epo (3 ng/mL) in a hypoxic chamber maintained with 2% O$_2$ for 5 days. Expanded cells were harvested and cultured in methylcellulose media (M3334 StemCell Technologies, Cambridge, MA, $1 \times 10^5$ expanded cells/mL) containing 3 U/mL Epo. Stress BFU-E was scored 5 days after culture.

## RNA-seq

HSCs from control or PHZ-treated mice were sorted into Trizol and RNA purified according to the manufacturer's instructions. DNase I-treated RNA samples were purified using the QIAGEN MinElute kit

before cDNA was made and amplified with a SMART-Seq2 protocol. The cDNA was then fragmented and barcoded for sequencing using a Nextera XT kit (Illumina, San Diego, CA). RNA-seq libraries were sequenced on an Illumina Sequencer (Illumina, San Diego, CA). Reads were mapped to mm10 using STAR (version 2.5.2b[85]), which was followed by differential expression analysis using DESeq2 (version 1.12.4[86]). GATA1-regulated genes in G1E-ER-GATA1 cells were extracted from ref. 58.

## scRNA-seq

Single cell RNA-seq libraries were prepared using the Chromium Next GEM Single Cell 3′ Kit v3.1 (10x Genomics, Pleasanton, CA) according to the manufacturer's instructions. Sorted LSK cells from the bone marrow and spleens of control or PHZ-treated mice were used to generate gel bead-in-emulsion (GEM). Libraries were sequenced by the Illumina NovaSeq platform (Illumina). Raw reads were aligned using Cell Ranger v3 pipeline, Cell Ranger Count (10x Genomics). After Counting reads, data were uploaded into R and analyzed with Seurat v4 software suite[87]. To measure cell doublets, each sample was prepared for DoubletFinder[88], by calculating mitochondrial percentage, normalized (Default). 2000 variable features were determined using VST, then scaled (ScaleData, Default Setting). For DoubletFinder, each sample PCA component was determined (RunPCA, Default), then dimensionally reduced using 10 PCA components (RunUMAP, Default). Cluster Resolution was set to 0.1. After samples were prepped, cell doublets were determined with following settings, PCs = 10, pN = 0.25, pK = 0.09, Doublet Expected rate = 10%. To merge samples and reduce batch effect, each sample was normalized using Seurat SCT transformation pipeline, regressing out mitochondria percentage, and variable features set to 16,000. Samples were merged using Seurat v4 integration pipeline. In total, 5000 integrating features were determined (SelectIntegratingFeatures), then samples were prepared for integration using the 5000 integration features (PrepSCTIntegration). Next Cells to anchor the four samples were determined (FindIntegrationAnchors) using 40 PCA dims, SCT normalization setting, and 5000 integrating features. Merged object was generated using IntegratingData using 40 PCA. Combined Object of all four samples was scaled, 40 PCA components was generated (RunPCA), Using Elbowplot and jackstraw, 20 PCA components was used for dimensional reduction (RunUMAP, min.dist=1). Optimal cluster resolution was determined manually to avoid over clustering.

## Bisulfite PCR and amplicon-seq

Bone marrow or splenic HSCs from control or PHZ-treated mice were sorted into PBS and subjected to bisulfite conversion using a EZ DNA Methylation-Direct Kits (Zymo Research, CA). Bisulfite converted gDNA were used for bisulfite PCR and PCR products were extracted from the gels using a Zymoclean Gel DNA Recovery Kit (Zymo Research, CA). Purified PCR products were used for amplicon-seq library generation using a Nextera XT kit. The library was sequenced on an Illumina iSeq 100 sequencing system. Sequences were mapped to mouse mm10 genome and analyzed with Bismark (v0.22.2). Alternatively, amplicons were cloned with a CloneJET PCR Cloning Kit (ThermoFisher Scientific) and individual colonies were sequenced.

## Quantification and statistical analysis

Statistical analyses were performed using GraphPad Prism 8.4.3 (GraphPad Software, San Diego, CA). To ensure the reproducibility of our findings, data were derived from multiple independent experiments performed on different days. Sample sizes were chosen based on observed effect sizes and standard errors from previous experiments, and data was checked for normality and similar variance between groups. Statistical details and sample number ($n$) of experiments are located in the figure legends. A $P$ value of 0.05 or lower was considered significant. Animal studies were performed without blinding.

## Reporting summary

Further information on research design is available in the Nature Portfolio Reporting Summary linked to this article.

## Data availability

The RNA sequencing dataset and bisulfite sequencing data generated in this study have been deposited in the Gene Expression Omnibus (GEO) under the accession numbers GSE182059 and GSE233845. Source data are provided with this paper.

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

## Acknowledgements

This work was supported by the National Institutes of Health (NIH, R01HL165145 and P01CA265748). D.N. is a Scholar of the Leukemia and Lymphoma Society. Y.K. was supported by the Uehara Memorial Foundation. R.L.M. and J.T. were supported by the NIH (F31DK112542 for R.L.M., F31HL164097 for J.T.). K.T. was supported by the Sabin Family Fellow Award, the ASH Scholar Award, the MD Anderson Physician Scientist Program, the MD Anderson AML/MDS Moonshot Program, and the NIH (P50CA100632 and R01CA237291). J.F.M. was supported by the NIH (HL127717 and HL118761), the Vivian L. Smith Foundation, and the MacDonald Research Fund Award (16RDM001). Flow cytometry was partially supported by the CPRIT Core Facility Support Award (RP180672) and the NIH (CA125123 and RR024574). Sequencing was partially supported by the NIH (P30CA125123). We thank Catherine Gillespie for editing the manuscript and Xiangguo Shi for assisting transplantation experiments.

## Author contributions

Contribution: Y.T. and D.N. conceptualized the research; Y.T., Y.K., and D.N. designed the experiments; Y.T. and Y.K. performed most of the experiments with help from R.L.M., A.K., J.T., and M.H.S.; Y.T., Y.K., and D.N. analyzed the results; R.L.M., J.H.K., K.A.H., H.U., K.T., and M.A.H.S. performed bioinformatics analysis; Y.T., Y.K., J.F.M., M.A.H.S., and D.N. wrote, edited and reviewed the manuscript; D.N. acquired the funding and administrated the project.

## Competing interests

J.F.M. is a co-founder of and owns shares in Yap Therapeutics. The remaining authors declare no competing interests.
