## [Peer Review File · Nature Communications]

Increased iron uptake by splenic hematopoietic stem cells promotes TET2-dependent erythroid regenerationREVIEWER COMMENTS

Reviewer #1 (Remarks to the Author):

The manuscript by Tseng et al. addresses the role of HSCs in stress erythropoiesis using a lineage-tracing approach and Tet2 mutant mice. A major conclusion was that splenic HSCs are responsible for stress-induced erythroid output. A recent series of studies have implicated progenitor cells in sustaining steady-state hematopoiesis in the mouse, apparently independent of HSCs. In addition, specialized stress-dependent erythroid progenitor cells have been linked to mediating the stress erythropoiesis response. Based on existing paradigms, if HSCs are activated by stress to produce large numbers of erythroid cells, this is not an obvious finding and has potential to be important. Overall, the manuscript is well-written, with the logic, data and details presented with high clarity. Recommendations are made below to address important unresolved issues and to further strengthen existing data.

Specific Comments:

1. The lineage-tracing system was described in a Blood Advances paper. However, I do not believe this system is not commonly used, and it is important to articulate more details to justify its utility in this specific context. Furthermore, as a major result of this study critically relies on this system, it would be important to consider alternative approaches to address the role of stress-activated HSCs versus (or in addition to) the established stress progenitor cells described by Paulson and colleagues (e.g., Chen et al. Blood 2020; Hao et al. Blood Adv. 2019; Xiang et al. Blood 2015).
2. While it is established that iron/heme promote erythropoiesis, it is not obvious that these components would act on HSCs to enhance erythroid output. It is important to consider alternative explanations, e.g. the iron/heme-dependent increase in erythroid locus priming in HSCs might be analogous to heme enhancing GATA1 function in erythroid precursor cells described by Bresnick and colleagues (e.g., Liao et al. Cell Reports 2020; Tanimura et al. Dev. Cell 2018). Although certain erythroid genes were affected, it would be important to assess whether these genes constitute a subset of the erythroid gene expression program, a major component of the program, or the full program.
3. Based on Tet2 intracellular flow cytometry and a qualitative immunofluorescence analysis, the authors concluded that erythroid stress increases Tet2 expression and activity in splenic HSCs. The flow analysis revealed an approximate 2 fold increase in immunoreactive protein with no significant change in mRNA. The qualitative IF analysis showing a single cell is difficult to interpret. Is there an alternative approach and/or better controls that can be incorporated into the current approach to yield a more definitive conclusion? As presented, the apparent protein, but not mRNA, change would suggest as post-transcriptional mechanism. Is this really correct?
4. Bisulfite sequencing at Tfrc in splenic HSCs suggested stress-dependent demethylation. Ideally, this analysis should be extended to genes implicated in promoting stress erythropoiesis e.g. selenoproteins (Trsp mutant) (Liao et al. Blood 2018), SpiC (Bennett et al. Science Sig 2019), Samd14 (Hewitt et al. Dev. Cell 2017) or others.

Reviewer #2 (Remarks to the Author):

In the present manuscript entitled, "Increased iron uptake by splenic hematopoietic stem cells promotes TET2-dependent erythroid regeneration," the authors demonstrated that splenic hematopoietic stem cells (HSCs) responded to anemia by increasing iron uptake in order to promote erythropoiesis via the epigenetic regulator TET2. Various cofactors, including iron and alpha-ketoglutarate, have been shown to regulate TET2 expression, providing intuitive support for the notion that TET2 may contribute to or be affected by anemia with low iron levels. Previously, Inokura et al. (2017) showed that knockdown of Tet2 led to the development of normocytic anemia, elevated serum levels of ferritin, increased mitochondrial ferritin in erythroblasts, and dysregulation of genes involved in iron and heme metabolism (Inokura et al. 2017). Further, TET2 expression increases in response to oxidative stress, and DNA demethylation via TET2 facilitates the expression of ferroportin and erythroferrone (Guo et al., 2017). It has also been shown that loss of TET2 leads to hyperproliferation and impaired differentiation of human colony-forming unit-erythroid cells (Qu et al., 2018; Yan et al., 2017). The authors expanded upon these studies by demonstrating that extramedullary splenic HSCs play a role in responding to acute anemia in a TET2-dependent fashion and by implicating iron uptake in HSCs in this mechanism. Strengths of this study include the Krt18-CreER labeling system, careful functional analysis of HSCs, and the measurement of iron in HSCs. The Krt18-CreER labeling system enables the tracing of HSCs and their progeny. The authors very carefully demonstrate both phenotypic and functional effects of acute, chemically-induced anemia on HSCs both in vitro and in vivo. They further showed that loss of Tet2 reduced the ability of splenic HSCs to generate erythroid colonies. Similar results to the colony-forming assays have been reported previously in the context of Tet2 deficiency (Qu et al., 2018). In general, the manuscript is also clear and well-written with some grammatical errors. It would be helpful to provide additional discussion of the phenylhydrazine (PHZ) system, the significance of CD169+ macrophages, and previous studies examining the role of TET2 in erythroid cells (as described above). The finding that splenic HSCs were uniquely affected in this model of acute anemia is very compelling and may be worth pursuing. The measurement of iron in HSCs and the in vitro studies culturing HSCs with iron and ascorbate appear to be somewhat novel and open up the potential for a new area examining the roles of ions and cofactors in hematopoietic stem and progenitor cells.

Weaknesses of this study include the use of PHZ and heat-stressed erythrocytes to induce anemia and lack of mechanistic elements. PHZ, a hydrazine derivative that is toxic to red blood cells, induces an acute form of anemia. While PHZ has been used previously, both the PHZ and heat-stressed erythrocytes used in this study seem to be somewhat artificial systems, and their biological relevance is unclear. PHZ does seem to induce a transient anemia, which may be beneficial in some experimental contexts; however, the significance of the study would be significantly enhanced by examining the role of TET2 in anemia in more biologically-relevant models. Consistently, the authors raise some concerns about whether the extramedullary hematopoiesis observed in their model translates to human anemic conditions. It is also possible that PHZ may have toxic effects on other tissues that could confound the results of the study. Given this possibility, the authors should demonstrate whether toxicity is observed, specifically in HSPCs. Furthermore, there are several different types of anemia observed in patients, including iron-deficiency anemia and anemia of chronic disease. This study would be further strengthened by demonstrating that the TET2-dependent mechanism that they described is detected in patients with anemia and whether it is universally observed in anemia or whether it is specific to certain types of anemia. This additional information would enhance the translational relevance of the study. Previous studies indicate that mutations in Tet2 may be particularly significant in the context of

myelodysplastic syndrome (MDS). As discussed above, there are some previous studies that have already established a role for TET2 in anemia, somewhat reducing the novelty of the present study. Consistent with other studies highlighting the role of cofactors in the regulation of TET2 expression, the authors note that TET2 expression is dependent on iron. While they demonstrate that suppression of iron levels or TET2 expression reduced the expression of erythroid genes and, conversely, that iron supplementation rescues this phenotype, the mechanisms underlying these processes are not fully explored. The authors mention that they observed an increase in splenic CD169+ macrophages, which they contend is consistent with the response to anemia; however, the significance of this observation and its relevance to the disease mechanism are unclear. The authors note that while TET2 was necessary for enhanced erythroid differentiation, it was not required for expansion of splenic HSCs in the context of anemia. This result is surprising as Tet2 mutations are frequently observed in patients with clonal hematopoiesis of indeterminate potential (CHIP), which is characterized by clonal expansion and as these mutations have been implicated as early events in the development of leukemogenesis. Further clarification of this result would be informative.

The innovation of this study is the role of HSCs in stress erythropoiesis in response to acute anemia and the connection of the TET2-dependent mechanism underlying this process with iron uptake in HSCs. This study provides additional support for a role for TET2 in non-myeloid hematopoietic cells, such as erythroid lineages. The significance of this study is that TET2 may serve as a therapeutic target to help stimulate erythropoiesis in the context of anemia. The contributions of Tet2 deficiency to anemia could have important clinical implications for patient stratification and treatment in hematological malignancies, including MDS, leukemia, and CHIP; however, the clinical significance of this study is not fully conveyed in the manuscript. The authors indicate that their results support a role for TET2 as a therapeutic target in the context of anemia, but it remains unclear how TET2 would be targeted, especially as the present study suggests that TET2 expression facilitates the response to anemia. In fact, loss-of-function mutations in TET2 have been observed in MDS patients with anemia. It is recommended that the authors consider including a more biologically-relevant model of anemia, more carefully examining the significance of their results in the context of previous studies, and more thoroughly characterizing the mechanism by which TET2 loss contributes to anemia.

Inokura et al., 2017 <https://pubmed.ncbi.nlm.nih.gov/28167288/>

Guo et al., 2017 <https://pubmed.ncbi.nlm.nih.gov/28697999/>

Qu et al., 2018 <https://pubmed.ncbi.nlm.nih.gov/30254129/>

Yan et al., 2017 <https://pubmed.ncbi.nlm.nih.gov/28167661/>

Reviewer #3 (Remarks to the Author):

Tseng et al present a series of experiments to address the mechanism by which the hematopoietic system in mice responds to phenylhydrazine-induced hemolytic anemia, a well-studied model for stress erythropoiesis. The new information the authors provide regarding this model is that 1) it appears that splenic HSCs are increased; it had previously been demonstrated that BFU-E and MPPs are increased but apparently HSC increases had

not been assessed; 2) that iron plays a role in stimulating the expansion and erythroid-specific differentiation of HSCs; 3) Tet2 plays a role in promoting the erythroid maturation but not the HSC expansion.

The basic conclusions from the experiments are sound, and the data provide new insight into hemolytic anemia. It would be of greater impact if the authors could "connect the dots" between iron, TET2, and splenic HSC expansion: as the work stands, these findings seem to be separated in their own silos. As noted below, the relationship of this work to that done by Paulson's group on the flexed-tail mouse is of considerable interest.

One major problem with this study is the question of its relevance to humans, since it is not clear that the human spleen plays any role in red cell production under any circumstances, including hemolytic anemia.

One way to assess the role of TET2 in human hemolytic anemia is to use clinical data from patients with TET2-mutated hematopoiesis (clonal hematopoiesis of indeterminate potential, seen in the elderly); TET2 is frequently one of the three genes mutated, along with ASXL1 and DNMT3A, in people without any hematopoietic abnormalities. Do TET2-mutated cells in these patients respond aberrantly to hemolytic anemias?

Some of the findings here are not dissimilar to those reported years ago by Paulson and co-workers (and others) on stress erythropoiesis, particularly the finding that the number of erythroid-biased colony forming units increases in the spleen following PHZ treatment. While it is appreciated that the data here are slightly different, the conclusions are very similar, if not essentially the same. Thus, it would be appreciated to provide the reader with informative referencing to the earlier work when presenting your findings, and to point out in more explicit/specific terms how the findings here differ (what assays had been used previously, and how they differ from the assays used here). Has no one previously examined HSC increases?

Some of the experiments are not sufficiently explained in the text to allow for appreciation/interpretation: in the HSC limiting dilution assay, what types of cells are detected in the peripheral blood of the recipients. And in Figure 4g – it is not stated at what time following PHZ the rbc number was assessed.

One point of interest is the necessity of TET2 after the HSC stage: Is TET2 expressed in erythroblasts? What would happen if the gene were deleted in EPO-R-positive cells?

Specific Points

1. The recovery from anemia (by day 5 following PHZ treatment) is more rapid than can be explained by the production of erythrocytes from HSCs. How is the increased erythrocyte number explained? Is there a contraction in the volume of the vascular space? Are these rbc's released from the spleen? Is there accelerated maturation of committed erythroblasts?
2. Please provide weights for the spleens, in addition to the pictures provided in Supplemental Figure 1B. The weight appears to increase 8-fold; does the cell number increase as well, and is this due to in situ proliferation or increased capture/trapping of cells from the peripheral blood? Can the increase in weight and cellularity be explained by the expansion of splenic HSCs (and their progeny) that the paper describes?
3. Exactly what is displayed in Figure 2b is not clear – is this the percent of HSCs that have clonogenic potential? If so, then the Y axis should be labeled as such.
4. Supplemental Figure D, E, F, and G is missing an important control, which is the

transfusion of non-heat-treated erythrocytes, to determine if the effects seen are merely due to transfusion itself, or due to the shortened life-span of the erythrocytes.

5. It is not clear why the experiments displayed in Figure 4 c and d utilized a transplantation model rather than analyzing the PHZ-treated mice themselves. Also, it appears that there was no marker allele, such as Rosa26-LSL-YFP to mark cells that had successfully undergone Cre-mediated collapse of the floxed allele via poly-dIdC injection. In the transplant recipient, how were collapsed/Tet2-null cells distinguished from non-collapsed/wildtype erythroblasts? It is not clear what the genotype of the recipient is, and, if it differs from donor in CD45 isotype, can this be used to distinguish donor from recipient erythroblasts? Usually, UBC-GFP is used as a marker to distinguish donor from recipient erythroblasts. While the data look interesting and significant, these questions need clarification.

6. Does Figure 4e display analysis of splenocytes taken from the PHZ-treated mice or from the spleens of transplanted recipient mice? It is not clear. Also, were sorted HSCs plated in the colony forming assay, or did you plate total splenocytes/bone marrow. The text states that the assay examines the “ability of the splenic HSCs to form erythroid colonies”.

7. Regarding Figure 4f: CFU-E is a descriptor based on a functional assay (colony formation) while pro-erythroblast is based on morphology, yet the data were attained based on flow cytometry. Please provide a reference that demonstrates the appropriateness of this.

8. In Figure 4g, it is not indicated at what time point following PHZ treatment that the measurements were made. In the text, it is stated that the recovery from hemolytic anemia was “delayed”, but figure 4g just provides a snapshot of one time point, so this “delay” is not evident.

9. In Figure 5a,b, c, the expression of TET2 is assessed at the RNA and protein level in HSCs. However, there could also be important differences in the erythroid precursors/erythroblasts. Was this assessed? Is there TET2 expression in erythroblasts? It does not appear this was addressed.

10. The failure of ascorbate to correct in Tet2-deficient HSCs is confusing since in other studies (Cimmino et al, ref in paper), ascorbate was able to correct stem cell defects in tet2-deficient mice. Also confusing is their conclusions from the Fig5f experiment: that it is the ferrous ammonium sulfate and not the ascorbate that is promoting tet2 function.

11. In the RNAseq analysis, it is not clear why gene expression in the PHZ-treated splenic HSCs is not compared with untreated splenic HSCs, rather than that of the bone marrow HSCs.

12. It is stated that the loss of tet2 significantly attenuated the induction of a number of erythropoiesis/iron-related genes. What percent of the induced genes were tet2-dependent and what percent were not?

13. A more global analysis of the changes in DNA methylation, and the dependency of tet2, during PHZ-induced hemolytic anemia should be provided.

14. In the discussion the authors state that TET2 “governs” the process, but that has not been demonstrated: it is certainly involved, but whether it alone can drive the process has not been shown.

15. How does TET2 dovetail with the need for BMP4 and Hedgehog signaling pathways? It has been shown by Paulson’s group that these two factors are critical for stress erythropoiesis. Are TET2-null cells able to respond to Shh and BMP4?

Minor points:

1. Figure 1d is introduced/discussed before Figure 1c.

2. TAM administration by gavage results in more rapid and potent activation of CreER.

3. Figure S4B is oddly presented and a bit difficult to interpret, since there are no markers/ticks on the side of the image for the different PCR products, the lanes are running

in a skewed manner, and there are no un-deleted (control) colonies from floxed mice.

4. Too many decimal points are provided for p values.

5. Please state in the methods how HSCs were isolated (what markers were used) for the immunofluorescence staining. Also provide more details on how the cells were sorted directly onto glass slides (equipment/instrumentation, used). Also, please provide details on how the quantitation of fluorescence was performed.

6. The figure legend for supplementary figure 5 is inadequate, as it does not explain what the color scheme indicates, and there is insufficient description of how these data/figures were generated – was the RNAseq data compared against previously obtained archival expression data that is publicly available?

REVIEWER COMMENTS

Reviewer #1 (Remarks to the Author):

The manuscript by Tseng et al. addresses the role of HSCs in stress erythropoiesis using a lineage-tracing approach and Tet2 mutant mice. A major conclusion was that splenic HSCs are responsible for stress-induced erythroid output. A recent series of studies have implicated progenitor cells in sustaining steady-state hematopoiesis in the mouse, apparently independent of HSCs. In addition, specialized stress-dependent erythroid progenitor cells have been linked to mediating the stress erythropoiesis response. Based on existing paradigms, if HSCs are activated by stress to produce large numbers of erythroid cells, this is not an obvious finding and has potential to be important. Overall, the manuscript is well-written, with the logic, data and details presented with high clarity. Recommendations are made below to address important unresolved issues and to further strengthen existing data.

Specific Comments:

1. The lineage-tracing system was described in a Blood Advances paper. However, I do not believe this system is not commonly used, and it is important to articulate more details to justify its utility in this specific context. Furthermore, as a major result of this study critically relies on this system, it would be important to consider alternative approaches to address the role of stress-activated HSCs versus (or in addition to) the established stress progenitor cells described by Paulson and colleagues (e.g., Chen et al. Blood 2020; Hao et al. Blood Adv. 2019; Xiang et al. Blood 2015).

RESPONSE: We thank the reviewer for pointing out that the use of the lineage tracing model was not clear. We have revised the text and provided background information of the HSC lineage studies and the justification to use such model in our study (page 7).

Additionally, we expanded our study to investigate how stress erythroid progenitors (shown by Paulson and colleagues, Harandi et al. JCI 120:4507). We transplanted control or PHZ-treated splenic HSC and analyzed three stress erythroid progenitor populations described by Paulson and colleagues (population I (c-kit⁺ CD71^{-low} Ter119^{-low}), II (c-kit⁺ CD71^{hi} Ter119^{med}) and III (c-kit⁺ CD71^{-med} Ter119^{hi})) in recipients' bone marrow and spleens. We found that mice transplanted with PHZ-treated splenic HSCs exhibited higher reconstitution in stress erythroid fraction I compared to control, while fraction II and III showed a trend towards increased reconstitution. These results indicate that PHZ-treated splenic HSCs have higher potential in generating stress erythroid progenitor compared to control. These new results are shown in Supplementary Figure 2D.

Finally, we performed single cell RNA-seq analysis of the bone marrow and splenic HSPCs isolated from PHZ-treated and untreated mice (Figure 3). This analysis shows that PHZ treatment significantly increases the fraction of splenic erythroblasts (cluster EB in Figure 3) and subfraction of myeloerythroid progenitor cells (cluster SC2) that are both developmentally downstream of HSCs. These two fractions were either absent (EB) or scarcely detected in the bone marrow (both control and PHZ treated) or the control spleen, consistent with the stress erythroid progenitors that emerges in the spleens following hemolytic anemia, described by Paulson and colleagues.

2. While it is established that iron/heme promote erythropoiesis, it is not obvious that these

components would act on HSCs to enhance erythroid output. It is important to consider alternative explanations, e.g. the iron/heme-dependent increase in erythroid locus priming in HSCs might be analogous to heme enhancing GATA1 function in erythroid precursor cells described by Bresnick and colleagues (e.g., Liao et al. Cell Reports 2020; Tanimura et al. Dev. Cell 2018). Although certain erythroid genes were affected, it would be important to assess whether these genes constitute a subset of the erythroid gene expression program, a major component of the program, or the full program.

RESPONSE: We compared our RNA-seq results with those from Bresnick and colleagues (EMBO Rep. 17:249-65) to identify Gata1-regulated genes that are also affected by PHZ in HSCs. One caveat of this analysis is that the dataset from Bresnick and colleagues is from G1E-ER-GATA-1 cells that resemble normal proerythroblasts derived from GATA-1 null embryonic stem cells, whereas our RNA-seq dataset is from adult HSCs. We identified 4,658 genes that were significantly (\log_2 fold change >1 or <-1, $p < 0.05$) changed by Gata1 induction in G1E-ER-GATA1 cells and identified in our RNA-seq dataset. Only 45 genes out of 4,658 Gata1-regulated genes were significantly different between bone marrow HSCs and PHZ-treated spleen HSCs. 32 out of 45 genes are induced by Gata1 and increased in PHZ-treated spleen HSCs, whereas the remaining 13 genes are repressed by Gata1 and downregulated in PHZ-treated spleen HSCs. The Gata1-induced/PHZ-induced genes include established erythroid genes such as Tfr3, Add2, Cldn13, and Ermap, while Gata1-repressed/PHZ-repressed genes include Kit, Mpl, and Lmo2 known to regulate HSCs. This indicates that although only a subset of Gata1-regulated genes is affected by PHZ treatment in spleen HSCs, the directionality of Gata1 gene regulation and the effects of PHZ in HSCs is the same. These new data are shown in the new Supplementary Fig. 5E.

3. Based on Tet2 intracellular flow cytometry and a qualitative immunofluorescence analysis, the authors concluded that erythroid stress increases Tet2 expression and activity in splenic HSCs. The flow analysis revealed an approximate 2 fold increase in immunoreactive protein with no significant change in mRNA. The qualitative IF analysis showing a single cell is difficult to interpret. Is there an alternative approach and/or better controls that can be incorporated into the current approach to yield a more definitive conclusion? As presented, the apparent protein, but not mRNA, change would suggest as post-transcriptional mechanism. Is this really correct?

RESPONSE: The data showing that Tet2 protein level increases in splenic HSCs after PHZ treatment was done solely by immunofluorescence staining of purified HSCs, and not by intracellular flow cytometry. Although we only show one HSC per group as a representative example in Figure 6b, we quantified Tet2 protein levels and provided data obtained from up to 83 HSC images in Figure 6c.

Our data showing that Tet2 protein level is increased without accompanying increase in mRNA level is consistent with the idea that Tet2 protein is subject to stabilization by multiple mechanisms. It has been shown that Tet2 is degraded in a manner dependent on calpain (Cell Rep 6,278-284) and caspase (Nature 497: 122-126) during stem cell differentiation. Additionally, it has been shown that Tet2 is stabilized by binding to 14-3-3 proteins (Nature. 559:637-641, JBC. 295:1754-1766). Consistently, we found that treating HSCs with a calpain inhibitor but not a caspase inhibitor increased Tet2 protein, suggesting that Tet2 protein levels may be dynamically regulated post-transcriptionally. However, as we note in the results section, the in vivo relevance of this calpain-mediated TET2 stabilization requires future studies. These new results are shown in Supplemental Figure 5B.

4. Bisulfite sequencing at Tfr3 in splenic HSCs suggested stress-dependent demethylation.

Ideally, this analysis should be extended to genes implicated in promoting stress erythropoiesis e.g. selenoproteins (Trsp mutant) (Liao et al. Blood 2018), SpiC (Bennett et al. Science Sig 2019), Samd14 (Hewitt et al. Dev. Cell 2017) or others.

RESPONSE: We have attempted to address this question by whole-genome bisulfite sequencing (WGBS) of bone marrow and splenic LSK cells from control and PHZ treated mice. However, we stumbled upon a significant roadblock due to the limited numbers of splenic LSK cells from the spleens of untreated mice. We were able to isolate less than 1,000 LSK cells from a mouse spleen. This is far from enough to obtain high quality WGBS data to unequivocally call differentially methylated regions in genes involved in erythropoiesis. We hope the reviewer finds the study being significantly improved based on our response to 3 critiques raised, including single cell analysis of erythroid commitment in HSPCs and TET2 stabilization.

Reviewer #2 (Remarks to the Author):

In the present manuscript entitled, “Increased iron uptake by splenic hematopoietic stem cells promotes TET2-dependent erythroid regeneration,” the authors demonstrated that splenic hematopoietic stem cells (HSCs) responded to anemia by increasing iron uptake in order to promote erythropoiesis via the epigenetic regulator TET2. Various cofactors, including iron and alpha-ketoglutarate, have been shown to regulate TET2 expression, providing intuitive support for the notion that TET2 may contribute to or be affected by anemia with low iron levels. Previously, Inokura et al. (2017) showed that knockdown of Tet2 led to the development of normocytic anemia, elevated serum levels of ferritin, increased mitochondrial ferritin in erythroblasts, and dysregulation of genes involved in iron and heme metabolism (Inokura et al. 2017). Further, TET2 expression increases in response to oxidative stress, and DNA demethylation via TET2 facilitates the expression of ferroportin and erythroferrone (Guo et al., 2017). It has also been shown that loss of TET2 leads to hyperproliferation and impaired differentiation of human colony-forming unit-erythroid cells (Qu et al., 2018; Yan et al., 2017). The authors expanded upon these studies by demonstrating that extramedullary splenic HSCs play a role in responding to acute anemia in a TET2-dependent fashion and by implicating iron uptake in HSCs in this mechanism. Strengths of this study include the Krt18-CreER labeling system, careful functional analysis of HSCs, and the measurement of iron in HSCs. The Krt18-CreER labeling system enables the tracing of HSCs and their progeny. The authors very carefully demonstrate both phenotypic and functional effects of acute, chemically-induced anemia on HSCs both in vitro and in vivo. They further showed that loss of Tet2 reduced the ability of splenic HSCs to generate erythroid colonies. Similar results to the colony-forming assays have been reported previously in the context of Tet2 deficiency (Qu et al., 2018). In general, the manuscript is also clear and well-written with some grammatical errors. It would be helpful to provide additional discussion of the phenylhydrazine (PHZ) system, the significance of CD169+ macrophages, and previous studies examining the role of TET2 in erythroid cells (as described above). The finding that splenic HSCs were uniquely affected in this model of acute anemia is very compelling and may be worth pursuing. The measurement of iron in HSCs and the in vitro studies culturing HSCs with iron and ascorbate appear to be somewhat novel and open up the potential for a new area examining the roles of ions and cofactors in hematopoietic stem and progenitor cells.

RESPONSE: We thank the reviewer for your enthusiasm and pointing out the strength of our study. Below, we have separated the reviewer’s comment into separate topics and provide point-to-point responses to each comment.

Weaknesses of this study include the use of PHZ and heat-stressed erythrocytes to induce anemia and lack of mechanistic elements. PHZ, a hydrazine derivative that is toxic to red blood cells, induces an acute form of anemia. While PHZ has been used previously, both the PHZ and heat-stressed erythrocytes used in this study seem to be somewhat artificial systems, and their biological relevance is unclear. PHZ does seem to induce a transient anemia, which may be beneficial in some experimental contexts; however, the significance of the study would be significantly enhanced by examining the role of TET2 in anemia in more biologically-relevant models. Consistently, the authors raise some concerns about whether the extramedullary hematopoiesis observed in their model translates to human anemic conditions. It is also possible that PHZ may have toxic effects on other tissues that could confound the results of the study. Given this possibility, the authors should demonstrate whether toxicity is observed, specifically in HSPCs. Furthermore, there are several different types of anemia observed in patients, including iron-deficiency anemia and anemia of chronic disease.

RESPONSE: Regarding the toxicity of phenylhydrazine (PHZ) to HSPCs, we have provided data showing that HSC numbers in the bone marrow is not affected by PHZ treatment (Fig. 1f), HSCs isolated from PHZ-treated mice have similar colony forming capacity as untreated HSCs (Fig. 2b-d), and that they have similar long-term reconstitution capacity as untreated HSCs in a purified HSC transplantation setting (Supplementary Fig. 2c). These results indicate that PHZ has little, if any, direct toxicity to HSCs.

We agree with the reviewer that PHZ may have toxic effects on other tissues. In fact, PHZ is known to have liver and kidney toxicity, although these may be secondary to hemolysis. In order to assess HSPC response to a more physiological anemia-inducing stress, we bled the mice, according to a study published by Irv Weissman and colleagues (Stem Cells Dev. 16:707-717). In this study, the authors showed that bleeding activates HSPC (c-kit⁺ Thy1.1^{low} Sca-1⁺ Lin^{-low}) proliferation. We now provide data showing that bleeding increases the absolute number and iron content in splenic HSCs (CD150⁺ CD48⁺ LSK cells) (new Figure 4i-k). These changes can be rescued by transfusing fresh red blood cells but not PBS. These results reveal that a clinically relevant model of anemia caused by repetitive bleeding induces similar responses of HSCs to the PHZ-induced hemolytic anemia model.

This study would be further strengthened by demonstrating that the TET2-dependent mechanism that they described is detected in patients with anemia and whether is universally observed in anemia or whether it is specific to certain types of anemia. This additional information would enhance the translational relevance of the study. Previous studies indicate that mutations in Tet2 may be particularly significant in the context of myelodysplastic syndrome (MDS). As discussed above, there are some previous studies that have already established a role for TET2 in anemia, somewhat reducing the novelty of the present study.

RESPONSE: We thank the reviewer for commenting on the human relevance and the novelty of the study. We would like to point out three key novelty of our study.

1) Our study demonstrates that HSCs have the ability to respond to anemia and replenish the particular lineage (erythroid cells) lost to the injury. Others have shown that MPPs are the earliest progenitors that respond to anemia, but our study establishes that HSCs produces MPPs and other erythroid progenitors for regeneration.

2) We identify splenic HSCs as a unique pool of HSCs that become activated and increase their regenerative potential upon anemia. Little is known regarding the difference between bone

marrow and splenic HSCs. To our knowledge, our study is the first to show functional differences between bone marrow and splenic HSCs in response to stressors.

3) We establish that iron has instructive roles in directing HSCs for regeneration after anemia. Iron has been mostly considered as a negative regulator of HSC function, but our study shows that iron taken up by HSCs in physiological context has pro-regenerative functions.

Regarding the question of the TET2-dependent mechanism is detected in patients, while we appreciate the suggestion on studying MDS patients with TET2 mutations, we respectfully contend that these studies are beyond the scope of the current study. Several genomics studies have analyzed large numbers of MDS patients and associated the MDS genomics to clinical phenotypes (NEJM. 364:2496, Leukemia. 28:281, Nat Genet. 49:204). By reviewing these studies, we were not able to find conclusive data demonstrating the association of TET2 mutations with anemia. In fact, although anemia (Hb <8g/dl) was associated with high risk MDS (HR: 4.24), TET2 mutations were not (HR: 1.31) (Ogawa and colleagues. Leukemia. 28:241).

We also considered that data from TET2 mutant CH may show an association between TET2 mutations and anemia. A study analyzing 676 anemic patients ≥60 years old found a small increase in the incidence of CH in anemic individuals (46.6%) compared to non-anemic individuals (39.1%, age-matched) (Blood. 135:1161). However, this study did not find any associations between the most common CH mutations (DNMT3A, TET2, ASXL1) and anemia. In CH persons, a higher maximum VAF was also not associated with anemia (3.35% in anemic vs. 2.80% in non-anemic). Finally, the incidence of age-related anemia is much higher (up to 44% of men 85 years and older (Mayo Clin Proc. 69:730-5, J Am Geriatr Soc. 40:489-96, Am Fam Physician. 39:129-36)) than that of TET2 mutant CH cases (5-10% of CH cases carrying TET2 mutations and ~20% of elderly population exhibiting CH (depending on the sequencing technology and detection limit) (Jaiswal et al. NEJM. 371:2488)), clearly demonstrating the multifactorial cause of anemia. Given these issues, we have revised the discussion to point out the limitation of the study (line 409-411).

Consistent with other studies highlighting the role of cofactors in the regulation of TET2 expression, the authors note that TET2 expression is dependent on iron. While they demonstrate that suppression of iron levels or TET2 expression reduced the expression of erythroid genes and, conversely, that iron supplementation rescues this phenotype, the mechanisms underlying these processes are not fully explored.

RESPONSE: Our data showing that Tet2 protein level is increased without accompanying increase in mRNA level is consistent with the idea that Tet2 protein is subject to stabilization by multiple mechanisms. It has been shown that Tet2 is degraded in a manner dependent on calpain (Cell Rep 6,278-284) and caspase (Nature 497: 122–126) during stem cell differentiation. Additionally, it has been shown that Tet2 is stabilized by binding to 14-3-3 proteins (Nature. 559:637-641, JBC. 295:1754-1766). Consistently, we found that treating HSCs with a calpain inhibitor but not a caspase inhibitor increased Tet2 protein, suggesting that Tet2 protein levels may be dynamically regulated post-transcriptionally. However, as we note in the results section, the in vivo relevance of this calpain-mediated TET2 stabilization requires future studies. These new results are shown in Supplemental Figure 5B.

The authors mention that they observed an increase in splenic CD169+ macrophages, which they contend is consistent with the response to anemia; however, the significance of this observation and its relevance to the disease mechanism are unclear.

RESPONSE: Thank you for pointing out the issue with our macrophage data lacking relevance to the whole story. We agree that these data as it stands will likely be confusing to the readers, and therefore we removed these data from the manuscript.

The authors note that while TET2 was necessary for enhanced erythroid differentiation, it was not required for expansion of splenic HSCs in the context of anemia. This result is surprising as Tet2 mutations are frequently observed in patients with clonal hematopoiesis of indeterminate potential (CHIP), which is characterized by clonal expansion and as these mutations have been implicated as early events in the development of leukemogenesis. Further clarification of this result would be informative.

RESPONSE: We believe the reason why Tet2 deletion did not further expand HSCs in mice treated with PHZ is because we analyzed the mice shortly (within 3 weeks of poly IC injection) after deleting Tet2 with the Mx1-Cre system. Moran-Crusio et al. (Cancer Cell. 20:11-24) demonstrated that Tet2 deletion leads to progressive HSPC expansion, whereby LSK cells were significantly expanded in 20 weeks old mice (Vav-Cre; Tet2 fl/fl) but only slightly in 4-6 weeks old mice. Consistently, we observed a trend towards increased bone marrow and spleen HSCs 3 weeks after Tet2 deletion (Fig. 5a), but the small increase in spleen HSCs was superseded by the effect of PHZ.

The innovation of this study is the role of HSCs in stress erythropoiesis in response to acute anemia and the connection of the TET2-dependent mechanism underlying this process with iron uptake in HSCs. This study provides additional support for a role for TET2 in non-myeloid hematopoietic cells, such as erythroid lineages. The significance of this study is that TET2 may serve as a therapeutic target to help stimulate erythropoiesis in the context of anemia. The contributions of Tet2 deficiency to anemia could have important clinical implications for patient stratification and treatment in hematological malignancies, including MDS, leukemia, and CHIP; however, the clinical significance of this study is not fully conveyed in the manuscript. The authors indicate that their results support a role for TET2 as a therapeutic target in the context of anemia, but it remains unclear how TET2 would be targeted, especially as the present study suggests that TET2 expression facilitates the response to anemia. In fact, loss-of-function mutations in TET2 have been observed in MDS patients with anemia. It is recommended that the authors consider including a more biologically-relevant model of anemia, more carefully examining the significance of their results in the context of previous studies, and more thoroughly characterizing the mechanism by which TET2 loss contributes to anemia.

Inokura et al., 2017 <https://pubmed.ncbi.nlm.nih.gov/28167288/>

Guo et al., 2017 <https://pubmed.ncbi.nlm.nih.gov/28697999/>

Qu et al., 2018 <https://pubmed.ncbi.nlm.nih.gov/30254129/>

Yan et al., 2017 <https://pubmed.ncbi.nlm.nih.gov/28167661/>

RESPONSE: We appreciate the reviewer for pointing out several references studying the role of TET2 in murine and human erythropoiesis. We have included these papers in the discussion and examined the significance of our study in the context of a broader context and in the context of human relevance, as suggested. We have expanded the study by demonstrating that a physiologically relevant model of anemia induced by repetitive bleeding causes similar changes to splenic HSCs to the PHZ-induced anemia model. We have also downplayed our suggestion

that TET2 (activation or stabilization) may be a therapeutic target for anemia. Finally, we performed single cell RNA-seq analysis of the bone marrow and splenic HSPCs isolated from PHZ-treated and untreated mice (Figure 3). This analysis shows that PHZ treatment significantly increases the fraction of splenic erythroblasts (cluster EB in Figure 3) and subfraction of myeloerythroid progenitor cells (cluster SC2) that are both developmentally downstream of HSCs. These two fractions were either absent (EB) or scarcely detected in the bone marrow (both control and PHZ treated) or the control spleen, consistent with the stress erythroid progenitors that emerges in the spleens following hemolytic anemia, described by Paulson and colleagues.

Reviewer #3 (Remarks to the Author):

Tseng et al present a series of experiments to address the mechanism by which the hematopoietic system in mice responds to phenylhydrazine-induced hemolytic anemia, a well-studied model for stress erythropoiesis. The new information the authors provide regarding this model is that 1) it appears that splenic HSCs are increased; it had previously been demonstrated that BFU-E and MPPs are increased but apparently HSC increases had not been assessed; 2) that iron plays a role in stimulating the expansion and erythroid-specific differentiation of HSCs; 3) Tet2 plays a role in promoting the erythroid maturation but not the HSC expansion.

The basic conclusions from the experiments are sound, and the data provide new insight into hemolytic anemia. It would be of greater impact if the authors could "connect the dots" between iron, TET2, and splenic HSC expansion: as the work stands, these findings seem to be separated in their own silos. As noted below, the relationship of this work to that done by Paulson's group on the flexed-tail mouse is of considerable interest.

One major problem with this study is the question of its relevance to humans, since it is not clear that the human spleen plays any role in red cell production under any circumstances, including hemolytic anemia.

One way to assess the role of TET2 in human hemolytic anemia is to use clinical data from patients with TET2-mutated hematopoiesis (clonal hematopoiesis of indeterminate potential, seen in the elderly); TET2 is frequently one of the three genes mutated, along with ASXL1 and DNMT3A, in people without any hematopoietic abnormalities. Do TET2-mutated cells in these patients respond aberrantly to hemolytic anemias?

RESPONSE: We appreciate the reviewer for the comment on the human relevance of our study. We have discussed this issue in the manuscript (page 16-17) that while the relevance of extramedullary hematopoiesis, particularly in the spleen, to hemolytic anemia is somewhat unclear, a recent case study of 1,933 extramedullary hematopoiesis (EMH) found 309 cases without myeloproliferative neoplasm (MPN) (Blood Cancer Journal 8:119). Among these 309 cases, 24 and 22 were associated with hemolytic anemia and thalassemia, respectively. 22 of the 24 cases (92%) of hemolytic anemia-associated EMH involved the spleen, while 11 out of 22 cases (50%) of thalassemia-associated EMH involved the spleen. Looking overall (which includes EMH cases with AML or MDS), spleen was the most frequently involved EMH site, with 164 out of 309 cases involving the spleen. We have revised the discussion to provide the actual numbers described in this paper to clarify the degree to which the spleen is involved in non-MPN associated EMH. Additionally, a recent study using single cell transcriptomics and colony

forming assays demonstrated that human splenic HSPCs in patients with hereditary spherocytosis displaying splenomegaly due to chronic anemia exhibit increased erythroid colony forming capacity and erythroid gene priming at a single cell level (Blood. 139:3387-3401). The authors in this study concluded that spleen HSPCs contribute to the erythropoietic response in chronic anemia in humans. Thus, the ability to respond to anemia and become primed to the erythroid lineage appears to be a conserved feature of splenic HSCs.

Regarding the question of whether TET2-mutant CH accompany aberrant erythropoiesis, we reviewed large scale CH studies but were not able to identify a link between TET2 mutations and anemia. A recent study analyzing 676 anemic patients ≥ 60 years old found a small increase in the incidence of CH in anemic individuals (46.6%) compared to non-anemic individuals (39.1%, age-matched) (Blood. 135:1161). However, this study did not find any associations between the most common CH mutations (DNMT3A, TET2, ASXL1) and anemia. In CH persons, a higher maximum VAF was also not associated with anemia (3.35% in anemic vs. 2.80% in non-anemic). Finally, the incidence of age-related anemia is much higher (up to 44% of men 85 years and older (Mayo Clin Proc. 69:730-5, J Am Geriatr Soc. 40:489-96, Am Fam Physician. 39:129-36)) than that of TET2 mutant CH cases (5-10% of CH cases carrying TET2 mutations and $\sim 20\%$ of elderly population exhibiting CH (depending on the sequencing technology and detection limit) (Jaiswal et al. NEJM. 371:2488)), clearly demonstrating the multifactorial cause of anemia.

We have also reviewed genomics studies that analyzed large numbers of MDS patients and associated the MDS genomics to clinical phenotypes (NEJM. 364:2496, Leukemia. 28:281, Nat Genet. 49:204). By reviewing these studies, we were not able to find conclusive data demonstrating the association of TET2 mutations with anemia. In fact, although anemia (Hb $< 8\text{g/dl}$) was associated with high risk MDS (HR: 4.24), TET2 mutations were not (HR: 1.31) (Ogawa and colleagues. Leukemia. 28:241). Given these issues, we have revised the discussion to point out the limitation of the study (line 409-411).

Some of the findings here are not dissimilar to those reported years ago by Paulson and co-workers (and others) on stress erythropoiesis, particularly the finding that the number of erythroid-biased colony forming units increases in the spleen following PHZ treatment. While it is appreciated that the data here are slightly different, the conclusions are very similar, if not essentially the same. Thus, it would be appreciated to provide the reader with informative referencing to the earlier work when presenting your findings, and to point out in more explicit/specific terms how the findings here differ (what assays had been used previously, and how they differ from the assays used here). Has no one previously examined HSC increases?

RESPONSE: Thank you for your comment on providing more clear referencing to earlier work and articulating our new findings. As pointed by the reviewer, Paulson and colleagues have extensively studied stress erythroid progenitors and their role after acute anemia (Blood. 105:2741, Blood. 113:911, JCI. 120:4507, Blood. 125:1803, Blood. 132:2580). In early studies, they demonstrated that splenic megakaryocyte-erythroid progenitors (MEPs) are the stress erythroid progenitors that respond to BMP4-Smad5 pathway using the flexed-tail (f/f) mutant mice (Lenox et al. Blood.105:2741). Subsequent studies found that immature progenitor cells exist in the bone marrow that can generate stress erythroid progenitors (Perry et al. Blood. 113:911), which was later determined to be CD34+LSK cells (ST-HSC or MPP, depending on the nomenclature) (Harandi et al. JCI. 120:4507). In this study, the authors showed that CD34-LSK cells (that contains HSCs) were significantly inferior to CD34+LSK (ST-HSCs/MPPs) in generating stress erythroid progenitors upon transplantation, leading to the conclusion that ST-HSCs generate stress erythroid progenitors in the spleen (Harandi et al. JCI. 120:4507). Further

studies have largely focused on ST-HSCs/MPPs as the source of stress erythropoiesis (reviewed in Paulson et al. *Exp Hematol.* 89:43).

As we have discussed in the introduction, other studies including ours have noted increase in splenic HSCs during pregnancy, after bleeding, and in an EPO-transgenic mouse model of anemia (*Nature.* 527:466, *Nature.* 505:555, *Stem Cell Reports.* 10:1908, *Stem Cell Dev.* 16:707). Our study extends these studies to show that HSCs respond to hemolytic anemia in a lineage tracing model, that the splenic HSCs are activated (unlike bone marrow HSCs) upon hemolytic anemia and exhibit robust multi-lineage reconstitution capability as would be expected from HSCs and not MPPs, and that iron and TET2-mediated gene expression changes are in play.

In response to this comment, we have expanded the introduction (page 4) with additional references from the Paulson lab. We have also changed to a more specific terms (such as BFU-E instead of erythroid progenitors) whenever possible to better illuminate the differences between prior studies and our study, which has focused on how HSC function changes with anemia.

Some of the experiments are not sufficiently explained in the text to allow for appreciation/interpretation: in the HSC limiting dilution assay, what types of cells are detected in the peripheral blood of the recipients. And in Figure 4g – it is not stated at what time following PHZ the rbc number was assessed.

RESPONSE: We define “responding” as recipient mice showing more than 1% reconstitution in all 5 lineages; myeloid cells, B cells, T cells, red blood cells, and platelets. Figure 4g (now changed to Figure 5g) was analyzed at day 5 after PHZ treatment. We have revised the figure legend and methods to clarify these and other experimental setups.

One point of interest is the necessity of TET2 after the HSC stage: Is TET2 expressed in erythroblasts? What would happen if the gene were deleted in EPO-R-positive cells?

RESPONSE: We assessed TET2 protein expression in proE and CFU-E and found that TET2 protein was expressed in these cells under steady-state, and its level was further increased upon PHZ treatment in vivo, particularly in splenic HSCs (new Supplementary Fig. 5A). Deleting *Tet2* largely abolished the immunofluorescence signal, attesting to the specificity of this assay. Thus, TET2 protein level is increased not only in HSCs but also in EPO responsive erythroid progenitors after hemolytic anemia.

Specific Points

1. The recovery from anemia (by day 5 following PHZ treatment) is more rapid than can be explained by the production of erythrocytes from HSCs. How is the increased erythrocyte number explained? Is there a contraction in the volume of the vascular space? Are these rbc released from the spleen? Is there accelerated maturation of committed erythroblasts?

RESPONSE: We believe that erythroid progenitors (such as those published by Paulson and colleagues) are also activated by hemolytic anemia and provide rapid erythroid output, while activated splenic HSCs provide a slower but long-lasting erythroid output for full recovery. Prior studies from Paulson and colleagues have shown that stress erythroid progenitors in the spleen (referred to as BFU-E based on in vitro assays) (Lenox et al. *Blood.* 105:2741) are immediately expanded after hemolytic anemia. These splenic BFU-E proliferated faster than bone marrow BFU-E in vitro and are considered as the first line of defense against anemia to produce

erythrocytes, that will be released to the bloodstream. Consistent with this model, our new single cell RNA-seq analysis of bone marrow and spleens from control and PHZ treated mice (new Figure 3) shows significant expansion of erythroblast-like cells (cluster EB in Figure 3) and erythroid-biased progenitor cells (cluster SC2 in Figure 3).

They also showed that these BFU-E are depleted after acute anemia and that some immature progenitor cells replenish them (Perry et al. Blood. 113:911). Further study identified CD34+LSK cells (ST-HSCs/MPPs) as the population that migrates from the bone marrow to the spleen to regenerate BFU-E via several intermediate stress erythroid progenitor cells (Harandi et al. JCI. 120:4507). CD34+LSK cells are likely to contribute to the long-term maintenance of stress erythroid progenitors and replenishment of them in preparation of subsequent anemia stress. We are not in disagreement with this model, and postulate that not only CD34+LSK cells but also HSCs (CD150+CD48-LSK) migrate to the spleen and contribute to the long-term maintenance of the erythropoietic potential against hemolytic anemia.

2. Please provide weights for the spleens, in addition to the pictures provided in Supplemental Figure 1B. The weight appears to increase 8-fold; does the cell number increase as well, and is this due to in situ proliferation or increased capture/trapping of cells from the peripheral blood? Can the increase in weight and cellularity be explained by the expansion of splenic HSCs (and their progeny) that the paper describes?

RESPONSE: We now provide the spleen weights in Supplemental Figure 1B. The increase spleen size is most likely due to massive red blood cell production as well as destructive red blood cell removal (Biochem Biophys Res Commun. 82:1320-4, Hematologica. 102:1304-1313). It is very unlikely that the expansion of splenic HSCs contributes to the increased spleen weight (76mg to ~350mg), as the number of HSCs per spleens only increases from ~1000 to ~10,000.

3. Exactly what is displayed in Figure 2b is not clear – is this the percent of HSCs that have clonogenic potential? If so, then the Y axis should be labeled as such.

RESPONSE: The reviewer is correct that the axis refers to the percentage of HSCs that have clonogenic potential. We have revised the figure accordingly.

4. Supplemental Figure D, E, F, and G is missing an important control, which is the transfusion of non-heat-treated erythrocytes, to determine if the effects seen are merely due to transfusion itself, or due to the shortened life-span of the erythrocytes.

RESPONSE: We now provide control data (new Supplementary Fig. 3E-F) by transplanting fresh red blood cells. Neither iron levels nor HSC numbers were increased after transfusion of fresh red blood cells.

5. It is not clear why the experiments displayed in Figure 4 c and d utilized a transplantation model rather than analyzing the PHZ-treated mice themselves. Also, it appears that there was no marker allele, such as Rosa26-LSL-YFP to mark cells that had successfully undergone Cre-mediated collapse of the floxed allele via poly-dIdC injection. In the transplant recipient, how were collapsed/Tet2-null cells distinguished from non-collapsed/wildtype erythroblasts? It is not clear what the genotype of the recipient is, and, if it differs from donor in CD45 isotype, can this be used to distinguish donor from recipient erythroblasts? Usually, UBC-GFP is used as a marker to distinguish donor from recipient erythroblasts. While the data look interesting and significant, these questions need clarification.

RESPONSE: We decided to perform HSC transplantation assays in Figure 4c-d (now changed to Figure 5c) in order to assess the functional changes in HSCs by PHZ treatment and in the absence of Tet2. Without performing transplantation, any changes in the frequencies or numbers of erythroid progenitors may be due to erythroid progenitor cells self-renewing or non-HSC progenitor cells generating them.

Regarding the question of how donor- and recipient-derived cells are distinguished, we used the allelic variants of CD45, CD45.1 and CD45.2. This was possible because unlike enucleated erythrocytes that do not express CD45, erythroid progenitor cells used in this study (ProE, EryA, B, C, defined by Socolovsky and colleagues, Blood. 108:123-133) express CD45 in recipient mice. We do note that only a subfraction of EryC expresses CD45. Additionally, a study showed that EryC (Ter119-high CD71-/low FSC low) do not express CD45 in unmanipulated mice (Exp. Hematol. 86:53). Given that our ability to detect CD45 in EryC could be due to the specific experimental setting (transplantation) we used, we removed the data on EryC to avoid confusion. We thank the reviewer for raising this question.

6. Does Figure 4e display analysis of splenocytes taken from the PHZ-treated mice or from the spleens of transplanted recipient mice? It is not clear. Also, were sorted HSCs plated in the colony forming assay, or did you plate total splenocytes/bone marrow. The text states that the assay examines the “ability of the splenic HSCs to form erythroid colonies”.

RESPONSE: Figure 4e (now changed to Figure 5d) shows the ability of bone marrow (BM) or spleen (SP) HSCs taken from non-transplanted mice, with or without PHZ treatment, to form colonies with erythroid cells. We single-cell sorted bone marrow or spleen HSCs into the culture media. The figure legend is edited accordingly.

7. Regarding Figure 4f: CFU-E is a descriptor based on a functional assay (colony formation) while pro-erythroblast is based on morphology, yet the data were attained based on flow cytometry. Please provide a reference that demonstrates the appropriateness of this.

RESPONSE: The gating strategies are based on Pronk et al. Cell Stem Cell. 1:428 (for CFU-E) and Socolovsky et al. Blood. 98:3261 (for pro-erythroblast, ProE). Specifically, CFU-E is identified as lineage⁻ c-kit⁺ CD41⁻ CD16/32⁻ CD105⁺ Ter119⁻ cells, whereas ProE is identified as CD71⁺ Ter119^{low}. Isolation of EryA (Ter119^{high}CD71^{high}FSC^{high}), EryB (Ter119^{high}CD71^{high}FSC^{low}), and EryC (Ter119^{high}CD71^{low}FSC^{low}) is based on a study from Socolovsky and colleagues (Liu et al. Blood. 108:123). These references are added to the methods of the revised manuscript.

8. In Figure 4g, it is not indicated at what time point following PHZ treatment that the measurements were made. In the text, it is stated that the recovery from hemolytic anemia was “delayed”, but figure 4g just provides a snapshot of one time point, so this “delay” is not evident.

RESPONSE: The time point being analyzed in Figure 4g (now changed to Figure 5g) was at day 5 after PHZ treatment. We have added this information to the figure legend. We agree with the reviewer on the use of the term delayed and have revised the text to read “Tet2 deficiency also impaired the recovery...” (line 263).

9. In Figure 5a,b, c, the expression of TET2 is assessed at the RNA and protein level in HSCs. However, there could also be important differences in the erythroid precursors/erythroblasts. Was this assessed? Is there TET2 expression in erythroblasts? It does not appear this was addressed.

RESPONSE: We assessed TET2 protein expression in proE and CFU-E and found that TET2 protein was expressed in these cells under steady-state, and its level was further increased upon PHZ treatment in vivo, particularly in splenic HSCs (new Supplementary Fig. 5A). Deleting Tet2 largely abolished the immunofluorescence signal, attesting to the specificity of this assay. Thus, TET2 protein level is increased not only in HSCs but also in EPO responsive erythroid progenitors after hemolytic anemia.

10. The failure of ascorbate to correct in Tet2-deficient HSCs is confusing since in other studies (Cimmino et al, ref in paper), ascorbate was able to correct stem cell defects in tet2-deficient mice. Also confusing is their conclusions from the Fig5f experiment: that it is the ferrous ammonium sulfate and not the ascorbate that is promoting tet2 function.

RESPONSE: We confirmed the results from Cimmino et al. that the aberrant self-renewal of Tet2 KO HSPCs can be corrected by ascorbate. We cultured whole bone marrow cells from Tet2^{+/+} and Tet2^{-/-} (Mx1-Cre; Tet2 fl/fl mice treated with poly (I:C)) with or without ascorbate supplementation and serially passaged the cells for 4 times. As shown in the new Supplementary Figure 5C, ascorbate treatment significantly corrected the aberrant self-renewal of Tet2-deficient HSPCs, similarly to Cimmino et al. Thus, our results show that ascorbate, while being able to suppress the aberrant self-renewal of Tet2-deficient bone marrow HSPCs, fails to promote erythroid commitment of Tet2-deficient HSCs. This could be due to the differences in the culture conditions (methylcellulose media that facilitates differentiation used in Suppl. Fig. 5C vs. HSC media only containing SCF and TPO in Fig. 6d-e) or possibly because of the differences in which other TET enzymes (particularly TET3) compensate for the loss of TET2. As discussed by Cimmino et al, TET3 may compensate sufficiently for lack of TET2 to diminish aberrant self-renewal upon ascorbate treatment, as shown by the lack of response of Tet2^{-/-} Tet3 KD cells to ascorbate by Cimmino et al. This mechanism may not operate for erythroid commitment of HSCs. Further studies are needed to clarify this question.

To address the question of whether the effects we saw in Figure 5f was due to FAS or ascorbate, we treated HSCs with FAS, ascorbate, or the combination (New Supplementary Figure 5D). This analysis revealed that while both FAS and ascorbate has some effects on erythroid gene expression on its own, the combination had the most significant effects.

11. In the RNAseq analysis, it is not clear why gene expression in the PHZ-treated splenic HSCs is not compared with untreated splenic HSCs, rather than that of the bone marrow HSCs.

RESPONSE: We were not able to perform RNA-seq analyses on control splenic HSCs because these cells are exceedingly rare in the spleen of untreated mice. There are ~ 500 splenic HSCs during homeostasis (frequency of ~0.002%) but less than 100 HSCs per spleen can be sorted. This is probably because the extreme low frequency of splenic HSCs makes the sorting efficiency even lower. We note this limitation in the revised manuscript (line 304). While we were not able to perform RNA-seq with control splenic HSCs, we performed new qPCR experiments with control splenic HSCs and tested the gene expression changes that happens with PHZ treatment on a gene-by-gene basis (Figure 6f). The new results confirmed that PHZ induces the expression of erythroid genes in splenic HSCs comparison to untreated control splenic HSCs.

12. It is stated that the loss of tet2 significantly attenuated the induction of a number of erythropoiesis/iron-related genes. What percent of the induced genes were tet2-dependent and what percent were not?

RESPONSE: We identified 9 out of the 30 most differentially upregulated genes exhibited erythroid-biased expression (Supplementary Figure 6). 6 of the 9 genes (shown in Figure 7f) were significantly reduced by Tet2 deletion. Slc25a37 showed a trend toward reduced expression in Tet2 KO HSCs but was not statistically different. We edited the text to specify that 6 out of 9 genes (instead of “a number of”) were Tet2 dependent (line 322).

13. A more global analysis of the changes in DNA methylation, and the dependency of tet2, during PHZ-induced hemolytic anemia should be provided.

RESPONSE: We have attempted to address this question by whole-genome bisulfite sequencing (WGBS) of bone marrow and splenic LSK cells from control and PHZ treated mice. However, we stumbled upon a significant roadblock due to the limited numbers of splenic LSK cells from the spleens of untreated mice. We were able to isolate less than 1,000 LSK cells from a mouse spleen. This is far from enough to obtain high quality WGBS data to unequivocally call differentially methylated regions in genes involved in erythropoiesis. We hope the reviewer finds the study being significantly improved based on our response to other critiques raised.

14. In the discussion the authors state that TET2 “governs” the process, but that has not been demonstrated: it is certainly involved, but whether it alone can drive the process has not been shown.

RESPONSE: We have reworded this to “regulated” (line 347).

15. How does TET2 dovetail with the need for BMP4 and Hedgehog signaling pathways? It has been shown by Paulson’s group that these two factors are critical for stress erythropoiesis. Are TET2-null cells able to respond to Shh and BMP4?

RESPONSE: We cultured Tet2-deficient bone marrow cells using the BMP4-containing media described by Paulson and colleagues to evaluate whether the BMP4 and Hedgehog signaling pathways promote the proliferation of stress erythroid progenitor in a Tet2-dependent manner. We found that Tet2 knockout bone marrow cells exhibited significantly decreased stress BFU-E number compared to WT, indicating that the BMP4/Hedgehog signaling pathways depend on Tet2 to activate erythroid progenitors. These results are shown in new Figure 5f.

Minor points:

1. Figure 1d is introduced/discussed before Figure 1c.

RESPONSE: We placed Figure 1c before Figure 1d for aesthetic reasons, so that two similar figures (Figure 1c and 1e) will be placed adjacent and the panels on the right are aligned vertically.

2. TAM administration by gavage results in more rapid and potent activation of CreER.

RESPONSE: Thank you for your advice. We adopted the intraperitoneal injection protocol from our previous study (Chapple et al. Blood Adv. 2018) as it was approved by the IACUC in our animal protocol, but we have adopted the gavaging protocol for ongoing studies.

3. Figure S4B is oddly presented and a bit difficult to interpret, since there are no markers/ticks on the side of the image for the different PCR products, the lanes are running in a skewed manner, and there are no un-deleted (control) colonies from floxed mice.

RESPONSE: We now provide a revised figure with the fl/fl control used in a separate experiment.

4. Too many decimal points are provided for p values.

RESPONSE: We believe the reviewer is referring to the p values shown in Figure 1h. We are happy to adhere to Nature Communications style guide on decimal place if available, but we were only able to find this information as relevant; “Where relevant, provide exact values for both significant and non-significant P values”. We have revised the decimal place following this guideline from another journal; “In general, P values larger than 0.01 should be reported to two decimal places, and those between 0.01 and 0.001 to three decimal places; P values smaller than 0.001 should be reported as $P < 0.001$ ”.

5. Please state in the methods how HSCs were isolated (what markers were used) for the immunofluorescence staining. Also provide more details on how the cells were sorted directly onto glass slides (equipment/instrumentation, used). Also, please provide details on how the quantitation of fluorescence was performed.

RESPONSE: HSCs were isolated as CD150+ CD48-/low LSK cells using BD FACSAria II, as described in the “Flow cytometry and HSC isolation” section of Methods. We added this information to the “HSC immunofluorescent staining” section as well. We directly sorted HSCs on a dry glass slide by placing the slide on top of the instrument’s tube holder. As each droplet produced by FACSAria II (70 um nozzle) is 1.5-2 nano L, they quickly dry up and most cells maintain their morphology. The slides with sorted cells were immediately placed in ice cold methanol for fixation in the sorting room. We quantified the fluorescence intensity of individual cells using the ImageJ software. We subtracted background fluorescence signal by quantifying the fluorescence signal of areas without cells. The Methods section is edited to clarify these points.

6. The figure legend for supplementary figure 5 is inadequate, as it does not explain what the color scheme indicates, and there is insufficient description of how these data/figures were generated – was the RNAseq data compared against previously obtained archival expression data that is publicly available?

RESPONSE: The data shown in Supplementary Figure 5 was taken from the Gene Expression Commons published in PLoS One. 7:e40321 by Irv Weissman and colleagues. This study analyzed more than 10,000 microarray data (at the time of publication) as a reference to provide empirical estimation of the absolute gene expression level in 39 hematopoietic cell populations. We apologize that there was no explanation on the color scheme and revised the figure to show color bars representing the dynamic range of the gene expression.

REVIEWERS' COMMENTS

Reviewer #1 (Remarks to the Author):

The revisions have appropriately addressed my prior recommendations.

The only remaining comment relates to Fig. 7F. The authors refer to NFE2 in the text, but show a MAF motif. MAFK heterodimerizes with p45/NFE2 and sometimes binds DNA as homodimers. The typical NFE2 motif is TGAGTCA with some wobble in certain circumstances. I do not believe the motif that was shown for MAF is an NFE2 motif. This may require modification.

Reviewer #2 (Remarks to the Author):

Authors have adequately addressed all of my concerns.

Reviewer #3 (Remarks to the Author):

The authors have adequately addressed most all of my concerns expressed in the initial review. I have only one additional/persistent concern, regarding extramedullary hematopoiesis: Although it is true, EMH can be found microscopically in human tissues, it is not clear how functional it actually is in humans. Based on the number of cells identified, when compared to the population found in the hematopoietic bone marrow, the role of EMH in humans for producing functional circulating erythrocytes may be limited, even in disease states. If the authors can provide any data from the literature on the functionality of splenic EMH in humans, derived from the literature, that would be helpful.

REVIEWER COMMENTS

Reviewer #1 (Remarks to the Author):

The revisions have appropriately addressed my prior recommendations.

The only remaining comment relates to Fig. 7F. The authors refer to NFE2 in the text, but show a MAF motif. MAFK heterodimerizes with p45/NFE2 and sometimes binds DNA as homodimers. The typical NFE2 motif is TGAGTCA with some wobble in certain circumstances. I do not believe the motif that was shown for MAF is an NFE2 motif. This may require modification.

RESPONSE: We are grateful for the reviewer for pointing this out before publication. The matrix ID of this MAFK motif in JASPAR is PB0146.1 and this motif was verified in a protein binding microarray study in *Science* 324:1720-1723 (2009). Based on the study design, the recombinant MAFK used in this study is likely to be in the homodimer state, and certainly not in a heterodimeric form with NFE2. Thus, referring to NFE2 in this context is misleading and we have removed the mentioning of NFE2 and cited the *Science* paper regarding the identified MAFK motif.

Reviewer #2 (Remarks to the Author):

Authors have adequately addressed all of my concerns.

RESPONSE: We thank the reviewer for the constructive critiques.

Reviewer #3 (Remarks to the Author):

The authors have adequately addressed most all of my concerns expressed in the initial review. I have only one additional/persistent concern, regarding extramedullary hematopoiesis: Although it is true, EMH can be found microscopically in human tissues, it is not clear how functional it actually is in humans. Based on the number of cells identified, when compared to the population found in the hematopoietic bone marrow, the role of EMH in humans for producing functional circulating erythrocytes may be limited, even in disease states. If the authors can provide any data from the literature on the functionality of splenic EMH in humans, derived from the literature, that would be helpful.

RESPONSE: A recent study (ref. 73) explored human EMH by comparing spleen, PB, and mobilized PB to BM using scRNA-seq and functional assays. As expected, splenic HSPCs showed no evidence of substantial ongoing hematopoiesis at steady state but demonstrated increased proliferation during stress erythropoiesis. This was observed by analyzing splenic HSPCs from patients with hereditary spherocytosis displaying splenomegaly due to chronic hemolytic anemia. These data provide strong evidence of the contribution of splenic HSPCs to stress erythropoiesis in humans, within the limitation of technical challenges and ethical considerations. The question of the relative role of bone marrow and extramedullary sites in supporting erythropoiesis during stress is a difficult question, even using mouse models as it is challenging to specifically label HSPCs in the extramedullary sites. We now acknowledge this limited understanding of the relative erythroid contribution of bone marrow versus extramedullary sites.